# Transkingdom interactions between *Lactobacilli* and hepatic mitochondria attenuate western diet-induced diabetes

Richard R. Rodrigues [1,7], Manoj Gurung [2,7], Zhipeng Li[2,7], Manuel García-Jaramillo [3], Renee Greer[2], Christopher Gaulke[4], Franziska Bauchinger[5], Hyekyoung You[2], Jacob W. Pederson [2], Stephany Vasquez-Perez[2], Kimberly D. White [2], Briana Frink[2], Benjamin Philmus [1], Donald B. Jump[3], Giorgio Trinchieri [6], David Berry[5], Thomas J. Sharpton[4], Amiran Dzutsev[6], Andrey Morgun[1,8 ✉] & Natalia Shulzhenko [2,8 ✉]

Western diet (WD) is one of the major culprits of metabolic disease including type 2 diabetes (T2D) with gut microbiota playing an important role in modulating effects of the diet. Herein, we use a data-driven approach (Transkingdom Network analysis) to model host-microbiome interactions under WD to infer which members of microbiota contribute to the altered host metabolism. Interrogation of this network pointed to taxa with potential beneficial or harmful effects on host's metabolism. We then validate the functional role of the predicted bacteria in regulating metabolism and show that they act via different host pathways. Our gene expression and electron microscopy studies show that two species from *Lactobacillus* genus act upon mitochondria in the liver leading to the improvement of lipid metabolism. Metabolomics analyses revealed that reduced glutathione may mediate these effects. Our study identifies potential probiotic strains for T2D and provides important insights into mechanisms of their action.

[1] College of Pharmacy, Oregon State University, Corvallis, OR, USA. [2] Veterinary Medicine, Oregon State University, Corvallis, OR, USA. [3] College of Public Health and Human Sciences, Oregon State University, Corvallis, OR, USA. [4] College of Science, Oregon State University, Corvallis, OR, USA. [5] Department of Microbiology and Ecosystem Science, University of Vienna, Vienna, Austria. [6] Cancer and Inflammation Program, Center for Cancer Research, National Cancer Institute, National Institutes of Health, Bethesda, MD, USA. [7] These authors contributed equally: Richard R. Rodrigues, Manoj Gurung, Zhipeng Li. [8] These authors jointly supervised this work: Andrey Morgun, Natalia Shulzhenko. ✉email: andriy.morgun@oregonstate.edu; natalia.shulzhenko@oregonstate.edu

ncreasing evidence underscores the importance of the micro-biome in human metabolic health and disease[1]. One of the most prevalent metabolic diseases, type 2 diabetes (T2D), is now a global pandemic and the number of patients that will be diagnosed with this disease is expected to further increase over the next decade[2]. The so-called "western diet" (WD, a diet high in saturated fats and refined sugars) has been recognized as one of the major culprits of T2D with gut microbiota playing an important role in modulating effects of diet[3,4]. Thus, there is an urgent need to elucidate the contributions of gut microbiota to metabolic damages caused by WD and to identify preventive approaches for T2D.

On the one hand, it is believed that complex changes in the structure of gut microbial communities, resulting from interactions of hundreds of different microbes, also called dysbiosis, underlies metabolic harm to the host[5]. On the other hand, some reports claim that individual members of the microbial community changed by the diet might have a significant impact on the host[6]. Although these two points of view are not necessary mutually exclusive, it is still unclear which hypothesis is more credible[7].

Herein, we used a data-driven systems biology approach (Transkingdom Network analysis) to model host–microbe interactions under WD and to investigate whether individual members of microbiota and/or their interactions contribute to altered host metabolism induced by the WD. The interrogation of the Transkingdom Network pointed to individual microbes with potential causal effects on the host's lipid and glucose metabolism. Furthermore, the analysis also enabled inference of whether microbes might elicit beneficial or harmful effects on the host. In addition, we detected associations between the frequencies of these microbes and obesity in humans. We then validated the functional role of the predicted bacteria in regulating metabolism by supplementing mice with these microbes. Next, gene expression, electron microscopy, and multi-omics network pointed to a novel finding that these two *Lactobacilli* may act by boosting mitochondrial health in the liver leading to the improvement in hepatic lipid and systemic glucose metabolism. Finally, the metabolomics analysis revealed few metabolites (e.g., reduced glutathione; GSH) that may mediate the beneficial effects of probiotics.

## Results

**Transkingdom Network predicts beneficial and harmful microbes.** We started by inducing T2D-like metabolic disease in C57BL/6 mice by feeding them a WD, which prior work has found to yield murine phenotypes that mimic human T2D[8–10]. As expected, when compared with mice receiving a control (normal) diet, the mice fed the WD exhibited glucose intolerance and insulin resistance (Fig. 1a, Fig. S1). The observed phenotypic changes were consistent at 4 and 8 weeks, as well as between replicate experiments. These results align with previous studies showing metabolic changes in male C57BL/6 J mice-fed WD[9,10]. Concurrently, the gut (ileum and stool) microbial communities were altered because of diet (Fig. 1b). Although gut location explained the majority of the variation in the microbial communities as expected[11,12] we observed robust changes in microbiota associated with feeding WD[8,13]. Interestingly, the overall composition of the gut microbiota was similar at 4 and 8 weeks of WD (Supplementary Data 1a).

Previous studies showed associations between ecological properties of microbial community (e.g., Shannon diversity) and host metabolism[14,15]. Therefore, we analyzed the association between several community parameters (Supplementary Data 1b)

and host phenotypes altered by WD. However, analysis of data from two separate time points (4 and 8 weeks of WD) and microbiome results from intestinal and fecal samples did not find any correlations that showed significant associations in both independent experiments (Supplementary Data 1c). Thus, it does not seem that general dysbiosis explains metabolic alterations in this experimental system.

Next, we sought to identify specific microbes regulating metabolic parameters using a Transkingdom (TK) network approach; this approach has been successfully used to identify key microbiota associated with various disease states, including human disease[16,17]. Towards this end, we created a TK network by integrating microbial abundances with systemic measurements of host metabolic parameters changed by the WD (Fig. 1c, Supplementary Data 2). The TK-network contained 1009 edges between 226 nodes (6 metabolic parameters and 220 microbial operational taxonomic units (OTUs)). The node degree distribution of the TK-network followed the power law function (Fig. 1c), supporting that the TK-network captures a cross-regulatory nature of the gut microbiota and host phenotypic ecosystem as power law had been shown as a critical property of biological networks[18,19]. Thus, the TK-network provided an opportunity to infer microbes responsible for controlling the overall composition of the microbial community (i.e., keystone species) as well as those that may control host phenotypes.

To identify microbes that likely contribute to T2D-related systemic changes in metabolism, we calculated a network property, called bipartite betweenness centrality (BiBC) that measures the frequency with which a node connects other microbe and host nodes in the graph[20]. We then integrated BiBC scores of each OTU with the WD-induced changes in abundance of ileal microbiota. A microbe was considered to be potentially beneficial (T2D *improver*) if it had a high-BiBC score and a lower abundance in the ileum of WD-fed mice (Supplementary Data 3). Conversely, a microbe was considered to be potentially *harmful* (i.e., a T2D worsener) if it had a high-BiBC score and a higher abundance in the ileum of mice fed WD (Supplementary Data 4).

As a result of these analyses, we identified four OTUs predicted to regulate glucose metabolism, which corresponded with high similarity to four bacterial species *Lactobacillus johnsonii, Lactobacillus gasseri, Romboutsia ilealis,* and *Ruminococcus gnavus* (Figs. 1d, e; Supplementary Data 16). The first two microbes were considered potentially beneficial (i.e., T2D phenotype *improvers*). The other two (*R. ilealis* and *R. gnavus*) were predicted to be worseners. Notably, *R. gnavus* has been previously shown to be associated with obesity[21,22]. Overall, these results indicate that individual microbes and/or their interactions and not community level dysbiosis (Fig. 1, Supplementary Data 1) could be key players in T2D.

It was proposed that keystone species have significant influence on the rest of gut microbiota, also characterized by a high number of connections within a network[23,24]. Therefore, we asked whether microbes with characteristics of keystone species in our network are among microbes that are predicted to influence host metabolic parameters. Using an approach developed by Berry and Widder[24], we investigated the microbial network and found one microbe with the closest match to *Bacteroides pectinophilus*, with a prominent keystoneness score, followed by few other microbes that also might qualify as keystone species (Figs. 1d, e, Supplementary Data 5, Supplementary Data 16). Notably, the candidate microbes predicted to affect the host had a low keystoneness score, suggesting that microbes with potentially high effect on the host do not necessarily play a central role in regulating the microbial community (Fig. 1d, Supplementary Data 5).

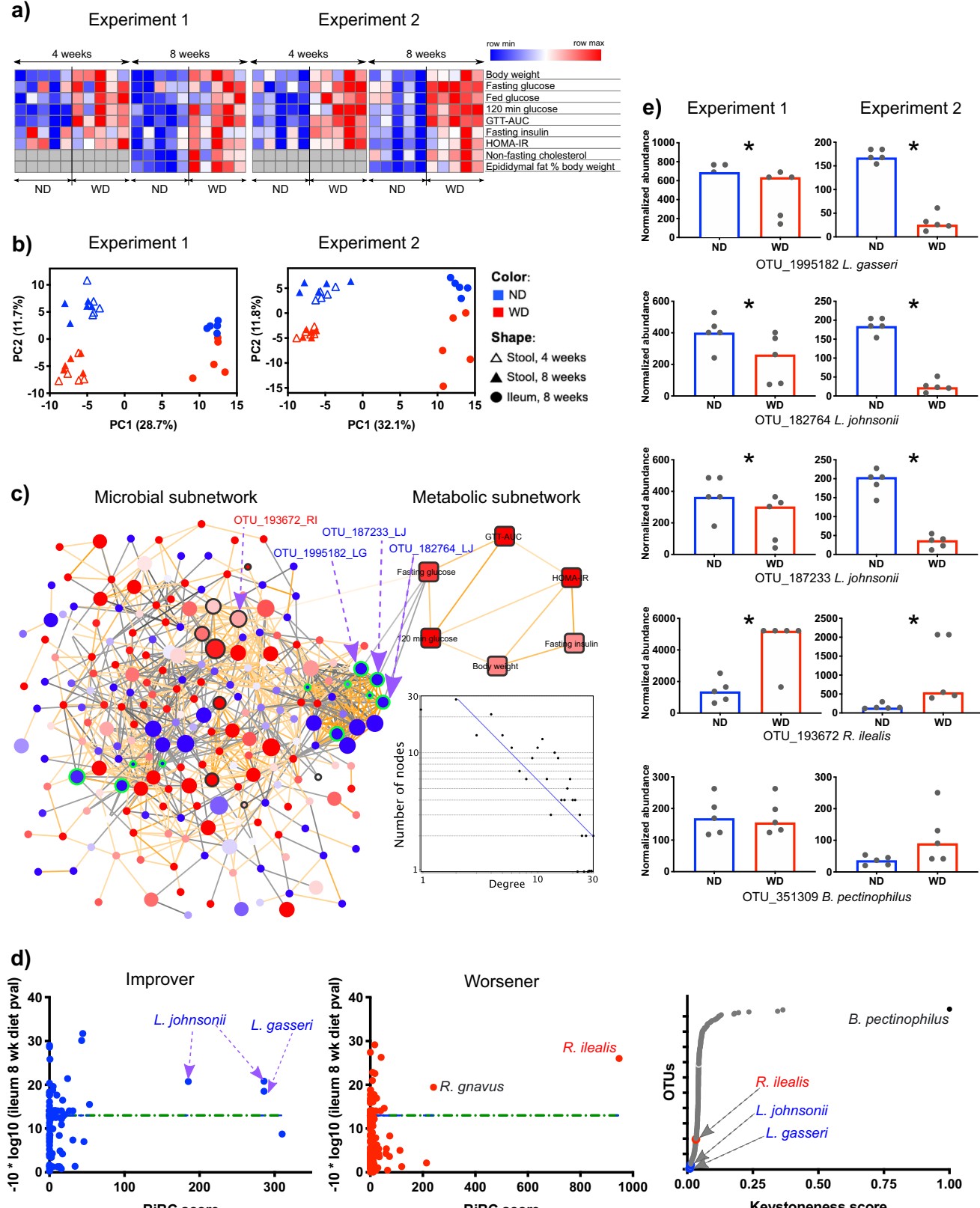

**Inferences from mice are validated by associations in humans.** To check the relevance of the candidate microbes in humans we identified a human study of a clinical population that consumes a WD-like diet and used the data to computationally evaluate our predictions[25]. In agreement with inferences from mouse data, we found correlations between body mass index (BMI) and the abundance of these microbial candidates (Fig. 2) in obese humans[25]. Specifically, the abundance of improvers was negatively correlated with BMI, whereas the abundance of the worsener was positively correlated. Furthermore, we found *R. ilealis* to be present in over 80% of obese patients, suggesting that this microbe could be a prevalent pathobiont in obese humans.

**Fig. 1 Inference of gut microbes affecting glucose metabolism in the host. a** The red and blue colors indicate higher and lower levels of metabolic parameters measured in mice fed normal diet (ND) or western diet (WD) at 4 and 8 weeks. Source data are provided as a Source Data file. **b** Principal Component Analysis of stool (triangle) and ileal (circle) microbial communities of mice on ND (blue) or WD (red). Source data are available at https://www.ncbi.nlm.nih.gov/sra/?term=PRJNA558801. **c** The microbe and host parameter nodes are represented by circles and squares, respectively, in the transkingdom (TK) network. Red and blue colors of nodes indicate increased and decreased (WD/ND) fold change, respectively, whereas the size of circle represents frequency of microbe in stool of WD mice. The black and green node borders indicate the microbes were significantly increased or decreased, respectively, in ileum of WD mice compared with ND (Fisher's $p$ value across experiments <0.05). The orange and black edges indicate positive and negative correlations, respectively. The degree distribution of the TK-network follows a power law. The blue line indicates the fitted line. Source data are available at https://tinyurl.com/TK-NW-Fig-1C. **d** The left two figures allow inference of microbial candidates that are potentially improvers (left figure) or worseners (middle figure) using high values of TK-network property (bipartite betweenness centrality (BiBC) on the $x$ axis) and significance of change in ileal (WD vs ND) abundance of microbes (log transformed Fisher's $p$ value across experiments on $y$ axis). The horizontal green line indicates a log transformed value for Fisher's $p$ value of 0.05. The right figure shows the keystoneness score ($x$ axis) of the microbial nodes ($y$ axis). Source data are provided as a Source Data file. **e** Ileal abundance of potential candidate and keystone microbes in ND and WD-fed mice at 8 weeks. Asterisk indicate the change in abundance passed statistical significance threshold (two-tail Mann–Whitney $p$ value <0.2 in each experiment, Fisher's $p$ value across experiments <0.05, and FDR < 10%. Each dot represents a mouse, bars present median of the group. Source data same as for **b**.

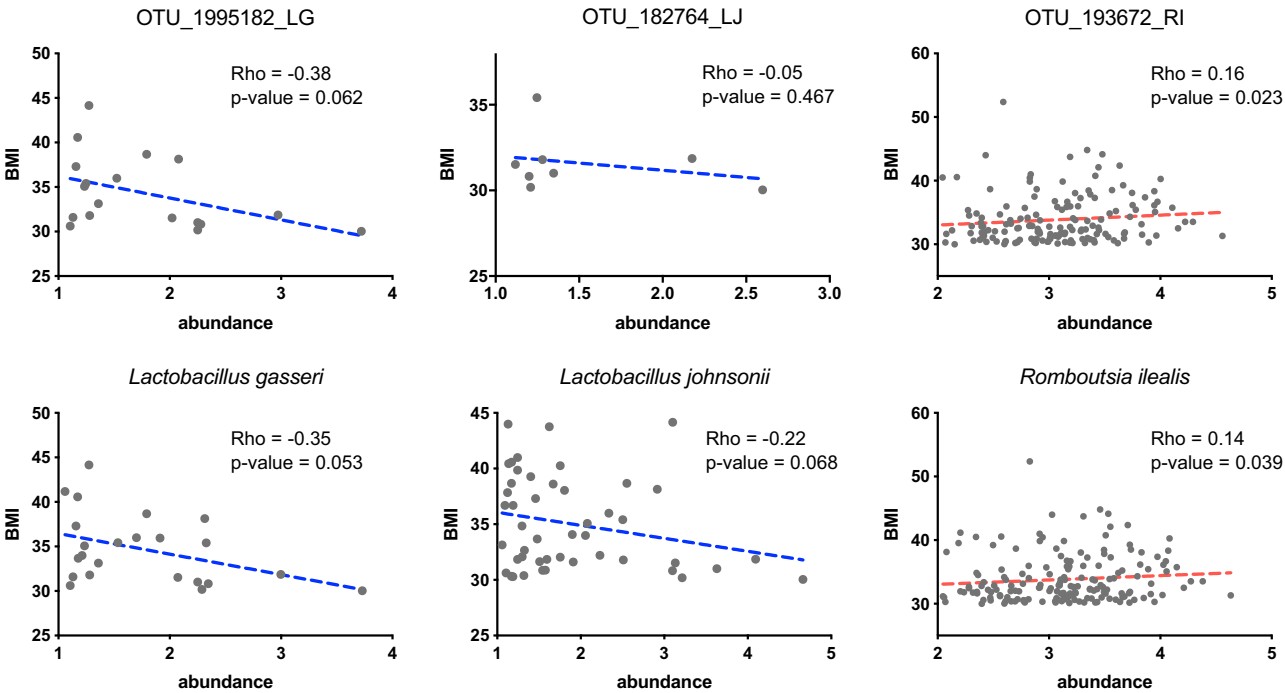

**Fig. 2 Computational verification of predicted microbes in human data from the literature[26].** Each scatterplot shows the abundance of the microbes ($X$ axis) in stool versus the BMI of obese humans ($Y$ axis). The dotted line indicates the fitted line. The Spearman rho correlation coefficient and one-tail $p$ value is shown. Data retrieved from www.ebi.ac.uk/metagenomics/studies/ERP015317.

Although the result for *R. ilealis* seemed to be more robust we observed only trend of positive association for *R. gnavus* that concurs with much smaller BiBC score for this bacterium (Figs. 1 and S2). Altogether, these observations provide further support for the predictions resulting from our analyses in the WD-fed mouse model.

**_Lactobacilli_ improve and _Romboutsia_ worsen glucose metabolism.** Encouraged by the support of our inferences in human data, we proceeded to test the role of *L. gasseri*, *L. johnsonii*, and *R. ilealis* in in vivo experiments designed according to predicted functional effects on the host. We anticipated that potential metabolic improvers (*L. gasseri*, *L. johnsonii*) would ameliorate metabolism damaged by WD, whereas the potential pathobiont (*R. ilealis*) would worsen metabolism in mice fed with normal diet. As predicted, WD-fed mice administered *L. gasseri* or *L. johnsonii* showed improved glucose tolerance (AUC and 120 min glucose levels) compared with mice on WD (Figs. 3a and S3). In

addition, supplementation with *L. gasseri* ameliorated the established glucose intolerance in mice (Figure S4). Conversely, mice supplemented with *R. ilealis* showed impaired glucose tolerance (15 mins. glucose levels in glucose tolerance test (GTT)) and reduced fasting insulin compared with mice fed with normal diet (Figs. 3a and S3). Accordingly, homeostatic model assessment (HOMA)-B, the index that reflects pancreatic beta-cell function, was also reduced by supplementation with *R. ilealis* (Fig. S3). These results suggest that the worsener/pathobiont and improver/probiotic microbes modulate the host systemic phenotypes likely via different mechanisms. Indeed, although higher levels of glucose early after glucose injection are most probably explained by decreased production of insulin in *R. ilealis* supplemented mice, *L. gasseri* and *L. johnsonii* improve glucose tolerance without altering insulin levels. Furthermore, whereas adiposity was not altered by *R. ilealis*, it was reduced in mice supplemented with improvers (*L. gasseri* or *L. johnsonii*) (Fig. 3a).

Although many human studies did not detect significant changes in fecal microbiota after probiotic administration[26–28],

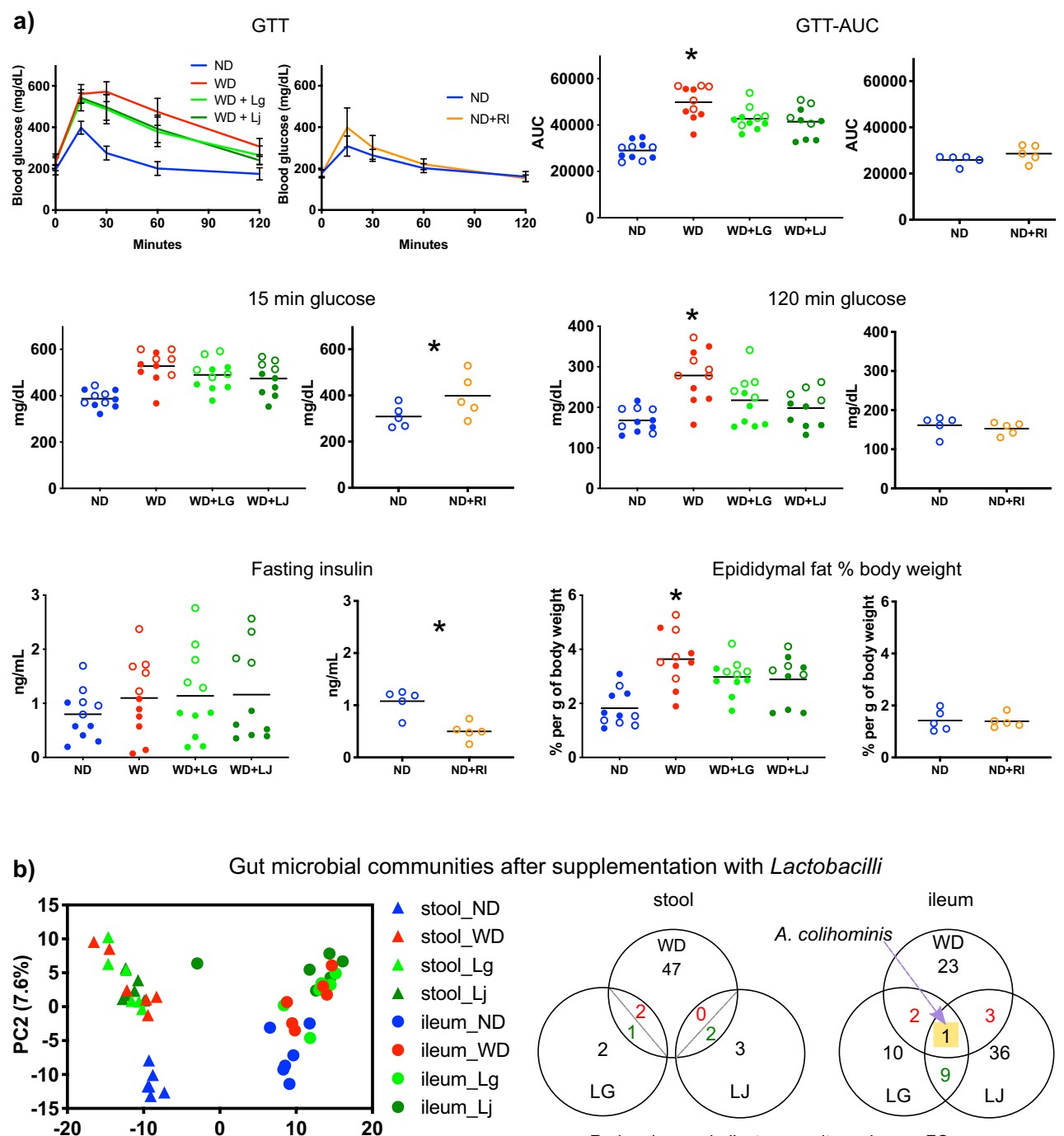

**Fig. 3 Experimental validation of microbial candidates. a** Metabolic parameters in mice given control diets and supplemented with or without the indicated microbe. Glucose tolerance test (GTT) curves show the mean and SD of blood glucose over time. Open and closed circles indicate two independent experiments; * indicates statistically significant differences in levels of the parameter between control group (WD for *Lactobacilli*, ND for *R. ilealis*) versus those supplemented with bacteria (one-tail *t* test *p* value <0.05 with FDR < 15%). Blue, ND; red, WD; light green WD with L. gasseri (WD + LG); dark green, WD with *L. johnsonii* (WD + LJ); orange, *R. ilealis* (ND + RI), respectively. Source data are provided as a Source Data file. **b** Principal Component Analysis of stool (triangle) and ileal (circle) microbial communities and Venn diagram of microbes changed in mice on ND, WD, WD + LG or WD + LJ and with >0.1% median abundance in at least one group across experiments (Fisher's *p* value <0.05 calculated using two-tail Mann–Whitney per experiment). For *Lactobacilli* supplementation experiments, $n = 11$ mice for ND, WD and WD + Lg groups, $n = 10$ mice for WD + Lj group. For *R. ilealis* (ND and ND + RI), $n = 5$ mice per group.

there were recent reports concerning the possible damaging effects of probiotics on the upper intestinal microbiota[29,30]. Therefore, we sequenced 16 S rRNA gene in ileum and fecal samples from mice supplemented with three candidate bacteria.

Very few changes were observed in the ileal and stool microbiota composition due to supplementation by these microbes (Fig. 3b, Fig. S5a, Supplementary Data 6). In hindsight, these results agree with the low keystoneness score of all three tested microbes that

have indicated their little influence on the rest of bacterial community (Fig. 1d). Furthermore, we did not find differences for individual taxa in stool samples in mice supplemented with bacteria. In the ileum, only one bacterium, *Anaerotruncus colihominis* (Supplementary Data 16), was reduced owing to western diet and increased by both *L. gasseri* and *L. johnsonii* (Fig. S5b). In agreement with our result, a study of gut microbiota from the Old Order Amish sect found this microbe to be negatively correlated with BMI and serum triglycerides[31]. Altogether, however, minimal alterations in microbiota induced by *L. gasseri* and *L. johnsonii* supplementation did not explain restoration of glucose metabolism promoted by these bacteria.

**Lactobacilli improve hepatic mitochondria and lipid metabolism.** Besides identifying effective probiotics for obesity/diabetes, it is critical to establish the host pathways through which these microbes exert their effect. Therefore, we next investigated two major target organs (intestine and liver) upon which both *Lactobacilli* might be acting to improve systemic metabolism. For a comprehensive evaluation of these organs we first analyzed global gene expression altered by *L. gasseri* and *L. johnsonii* supplementation. To identify common mechanisms by which *L. gasseri* and *L. johnsonii* improve metabolism, we focused on the genes that responded similarly to both microbes by identifying genes differentially expressed between both *L. gasseri* and *L. johnsonii* comparing with WD. The transcriptome of the ileum and liver showed distinct changes in response to supplementation by these bacteria (Fig. 4a). In striking contrast to the number of genes differentially expressed in the ileum (152, false discovery rate; FDR < 10%), there were much higher numbers of genes differentially expressed in the liver (654, FDR < 10%) (Supplementary Data 7–8). Furthermore, the great majority (638/654) of these genes were upregulated by *Lactobacilli* supplementation.

Functional enrichment analysis showed that genes that were changed in the ileum were enriched for only a few categories with the circadian rhythm function as the main one (Supplementary Data 9). Notably, one of the genes was Nfil3, which was downregulated in the ileum of *L. gasseri* or *L. johnsonii* supplemented mice as compared with the WD mice (Supplementary Data 7). In agreement with our results, the knockout of this gene in the intestinal epithelium had been shown to prevent mice from obesity, insulin resistance, and glucose intolerance[32].

Pathway enrichment analysis in liver, however, showed that multiple categories, and processes related to mitochondrial functions were over-represented among genes upregulated by *L. gasseri* and *L. johnsonii* (Figs. 4b and S6, Supplementary Data 10). In addition, further analysis demonstrated that genes belonging to all five mitochondrial complexes of the oxidative phosphorylation pathway (Fig. 4c) were upregulated in the liver of *L. gasseri* and *L. johnsonii* supplemented mice (Supplementary Data 8). There was also a group of genes coding for large and small subunits of mitochondrial ribosomal proteins with increased levels of expression in the *L. gasseri* and *L. johnsonii* group. Furthermore, genes involved in mitochondrial fusion were upregulated by the *Lactobacilli* including mitofusin 1 and 2 (Mfn1, Mfn2), mitoguardin 2 (Miga2), and optic atrophy 1 (Opa1) (Supplementary Data 8).

Hepatic mitochondrial functions are well known to be dysregulated in T2D[33–35]. Overall, our results suggest that in addition to mitochondrial functions, these probiotic bacteria induced structural/morphological changes in liver mitochondria. Thus, we performed electron microscopy of the livers from mice fed with WD and supplemented or not with each *Lactobacilli* (i.e., WD, WD + LG, WD + LJ) (Fig. 4d). Although there was no difference in the number of mitochondria, overall area occupied

by mitochondria was larger in WD group mice than in *L. gasseri* or *L. johnsonii* (Fig. 4e) suggesting increased size of mitochondria in livers of WD as compared with mice supplemented by *Lactobacilli*. This result indicates that mitochondrial swelling caused by WD, a phenomenon that can perturb proper functioning of mitochondria[36–38], was ameliorated by probiotic supplementation.

Next, we undertook quantitative evaluation of mitochondrial ultrastructural changes. Current agreement in the field is that healthy and damaged mitochondria correspond to dark, electron-dense and lucent, fragmented cristae images, respectively[37,39]. According to those criteria, we first identified a set of healthy and damaged mitochondria within individual images (Fig. 4f). Next, we estimated, in an unbiased manner (i.e., comparing healthy and good mitochondria within a given sample), which image parameters discriminated between the two types of mitochondria. We found lower values of standard deviation, integrated density, and the density mode in healthy compared with damaged mitochondria (note, in grayscale, white is 255 and black is 0) (Supplementary Data 11). Comparison between the three groups of mice showed significantly lower levels of these parameters in *L. gasseri* and *L. johnsonii* groups than in WD (Fig. 4f), pointing to healthier mitochondria in the former two groups of mice. Overall, these results support the prediction derived from gene expression data and indicate that *L. gasseri* and *L. johnsonii* supplementation prevented hepatic mitochondrial damage induced by western diet.

One of the important consequences of improved mitochondrial health is a restoration of fatty acid beta-oxidation. This process decreases build-up of detrimental fatty acids in the liver leading to improved systemic glucose metabolism[40,41]. In our data, among 19 regulated genes from the beta-oxidation gene subset, 18 genes were upregulated by supplementation of probiotic strains (Supplementary Data 12). Among upregulated genes were those involved in fatty acid transport (Slc25a17, Slc27a2), oxidation (Acads, Acadl) and hydration (Echs1) of fatty acyl representing major steps of beta-oxidation. These results pointed to possible increase in catabolism of fatty acids by *Lactobacilli* supplementation. Indeed, we found overall reduction of total hepatic lipids including several most abundant fatty acids known to have damaging effects on metabolism associated with T2D[42] such as monounsaturated fatty acids, oleic, and palmitic acids (Figs. 4g and S5c, Supplementary Data 13). Overall, these results are in accordance with the idea that changes in liver fat are central to development as well as reversion of T2D[43].

Besides fatty acids metabolism, two genes with well-established functions in cholesterol metabolism were also upregulated by both *Lactobacilli*: Abcg8, (hepatic cholesterol efflux[44]) and Cyp7a1, (conversion of cholesterol into bile acids[45]) (Fig. 4h). Therefore, we measured cholesterol in liver and serum samples. Although there was no change in serum cholesterol, there were reduced levels of liver total cholesterol in mice supplemented with *L. gasseri* or *L. johnsonii* (Fig. 4h). These results agree with an idea that alterations in the liver might precede lipid alterations detectable in serum[43].

**Multi-omic network infers key liver genes for effects of *Lactobacilli*.** To identify potential mechanisms by which *Lactobacilli* alter lipid and glucose metabolism, we created a multi-omic network by integrating the gene expression changed by *Lactobacilli* and lipid profile from the liver with systemic measurements of metabolic parameters changed by the WD (Fig. 5a). The multi-omic network contained 1776 edges connecting 380 nodes. The node degree distribution of this network followed the power law function (Figure S7), a critical property of biological networks[18,19]. Furthermore, although over half of differentially

expressed genes made into the multi-omic network, the enrichment analysis showed similar results with mitochondrial translation, fusion, organization, and autophagy formations being top enriched functions in this network (Fig. 5b). Next, we interrogated this network to infer genes regulated by *Lactobacilli* and potentially responsible for changing the systemic phenotypes. Specifically, we used the degree (local network property counting

the immediate neighbors) and BiBC[20], which is a global network property that measures the overall frequency with which a node connects to the nodes of other omics-type in the graph. Noteworthy, we found that gene expression nodes were predominantly connected to GTT, fasting glucose and 120 min glucose, two of which were significantly decreased by *Lactobacilli* supplementation (Fig. 5a–c). Furthermore, Ifitm3, Usp50, Rai12 (Elp5), and

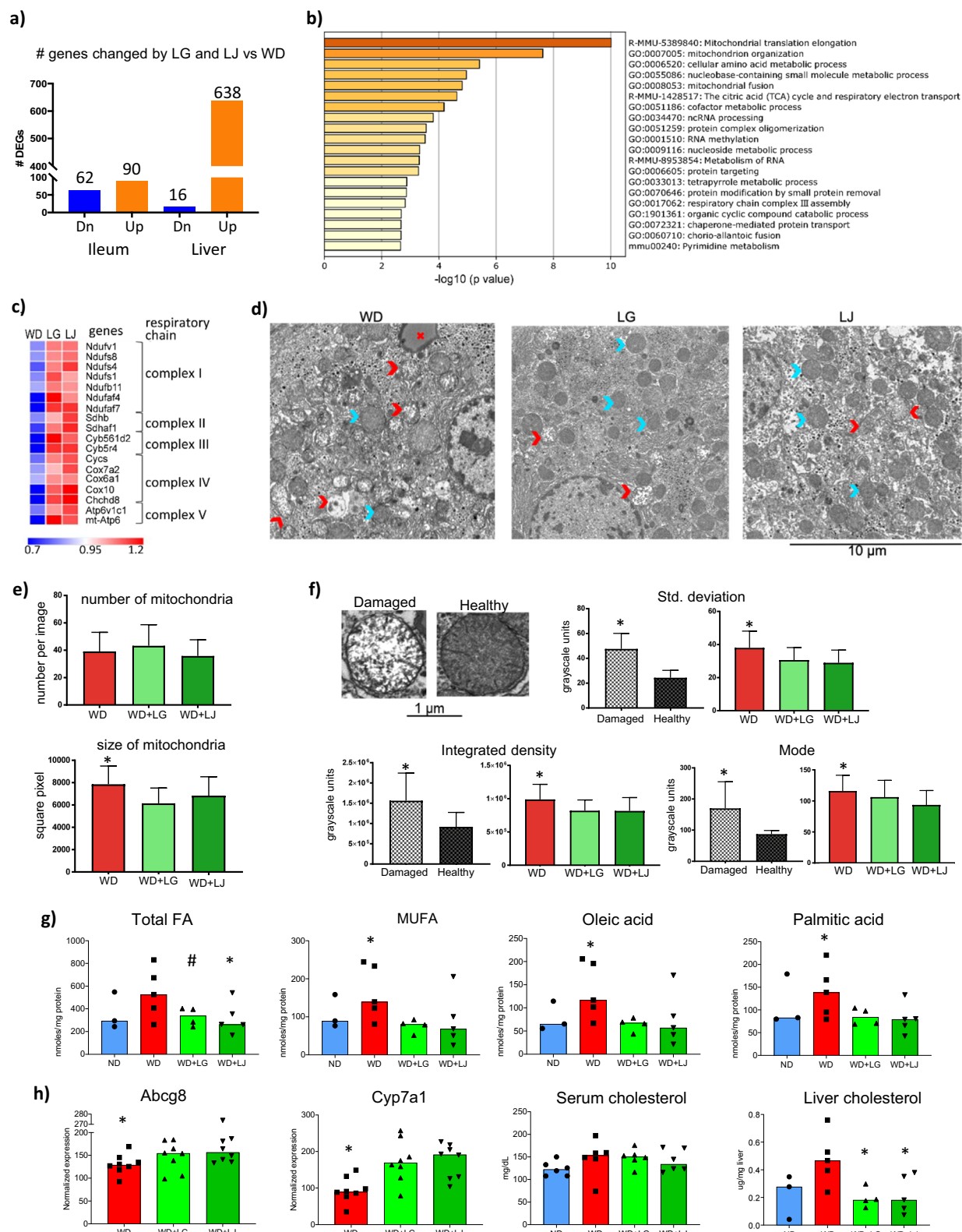

**Fig. 4 Transcriptome analysis, liver mitochondria, and lipids after supplementation with *L. gasseri* or *L. johnsonii*. a** Number of differently expressed genes (#DEGs, two-sided *t* test *p* value <5% in each Lactobacilli, Fisher's *p* value <5% calculated over both *Lactobacilli*, and FDR < 10%) regulated by *L. gasseri* and *L. johnsonii* in the same direction comparing to western diet. **b** Over-represented processes in the genes of the network shown in **a** of mice supplemented with *Lactobacilli*. **c** A heatmap showing the median expression of genes from the respiratory chain process in the livers of mice. **d** Representative electron microscope images of liver cells. The blue and red arrows indicate healthy and damaged mitochondria, respectively. **e, f** Various metrics of mitochondria in the liver of mice; *statistically significant differences between control and groups supplemented with bacteria (one-sided *t* test *p* value <5%). Data are presented as mean ± s.d. (*n* = 40 images for WD, *n* = 35 images for WD + LG and *n* = 37 images for WD + LJ groups; *n* = 60 mitochondria for healthy and *n* = 61 for damaged mitochondria). Source data are provided as a Source Data file. **g** Levels of long-chain fatty acids, **h** expression of cholesterol metabolism genes in livers, cholesterol levels in serum and liver of mice fed WD and supplemented with or without *Lactobacilli*. Each symbol represents one mouse, bars are median values. Source data are provided as a Source Data file; *n* = 3–5 mice per group (except serum cholesterol where *n* = 10–11 mice per group); * indicates statistically significant differences in WD vs WD + LG or LJ (one-sided *t* test *p* value <5%); # indicates *p* = 0.065.

Snap47, which are known to be involved in the maintenance of functional mitochondria[46–48], were found as key genes connecting expression alterations with systemic glucose metabolism (Fig. 5c). Interestingly, epididymal fat (also decreased in mice by *Lactobacilli*) was highly connected to liver fatty acids and to only one gene (Mfsd3), which codes for a solute carrier previously found in association with palmitic acid levels in a genome-wide association study[49].

Thus, the network analysis further suggested that the expression of genes responsible for mitochondrial organization and maintenance in the liver is the primary driver of improved systemic glucose metabolism.

**L. gasseri and L. johnsonii increase serum GSH and bilirubin.** Next, we applied a metabolomics approach to identify potential mechanisms responsible for improved hepatic mitochondrial health evoked by *Lactobacilli*. First, we established that metabolites were specifically increased by these bacteria in the serum of mice that did not contain other microbes. For this, germ-free mice fed WD were monocolonized or not with *L. gasseri* for 2 weeks and mouse serum was subject to metabolite profiling. Out of 133 metabolites that were identified (Supplementary Data 14a), 12 were increased after monocolonization, ranging from twofold for 8-iso-15-keto-PGF2a to 48 for bilirubin (Fig. 5d, Supplementary Data 14b). After this pre-selection in monocolonized mice, we compared abundance of the 12 metabolites between pools of sera of SPF mice supplemented with *L. gasseri* or *L. johnsonii* in three independent experiments (see details in Methods). We found that reduced (but not oxidized) GSH increased about four times, and bilirubin showed a trend to increased levels (FDR = 0.12), whereas two tauro-conjugated bile acids and 3-hydroxytetradecanedioic fatty acid showed various levels of decrease in *Lactobacilli* supplemented SPF mice (Fig. 5e, Supplementary Data 14c).

Although the mechanisms of GSH surge by *Lactobacilli* is not clear yet, this metabolite seemed to be a plausible candidate to cause hepatic mitochondrial improvement in mice as its antioxidant functions are well-established[50]. To test this hypothesis, we used AML-12 cell culture mimicking diabetic alterations in liver by adding high concentrations of fructose and glucose. Treatment of cells with different concentrations of GSH (in high sugar) enhanced expression of several genes with well-known mitochondrial functions such as mt-Atp6, Ndufv1, Mfn1, Opa1, Foxo3, Gabpa whose expression was also upregulated by *Lactobacilli* in the livers of mice (Fig. 5f, Supplementary Data 15a). We further tested three genes (Usp50, Ifitm3, Rai12) predicted by the network analysis (Fig. 5c) to play a key role in the control of mitochondrial health in liver and systemic glucose metabolism and have been previously shown to support mitochondrial homeostasis[47,48]. While we could not detect Usp50 in cell culture, the two other genes (Ifitm3, Rai12) showed increased expression in 6 and 9 mM GSH similar to other mitochondrial

genes (Fig. 5f). Thus, altogether these results indicate that an increase in GSH in the serum of mice is likely to be one of the important mechanisms used by *Lactobacilli* for boosting liver mitochondrial and antioxidant function, consequently improving systemic glucose metabolism.

## Discussion

Our work provides further support for the hypothesis that variations in abundance of a few key (but not keystone) microbes rather than overall changes of the microbial community might explain microbiota-related damage caused by western diet in T2D. Indeed, administration of two bacteria (*L. gasseri* and *L. johnsonii*), decreased by western diet, improved systemic glucose metabolism. The fact that this improvement could be achieved by supplementation of single bacteria, however, does not eliminate a possibility of microbe–microbe interaction playing a role in this process. Furthermore, both *Lactobacilli* had very low keystoneness, and accordingly we did not detect strong alterations in the gut microbiota (fecal or ileal) of mice supplemented by these two microbes. This is in agreement with several human studies that used other strains of probiotic bacteria and largely did not observe changes in taxonomic composition of fecal microbiota[26–28]. In contrast, two recent reports showed alterations in human mucosal microbiota communities by probiotics and potential adverse side effects of probiotics, especially when used after antibiotics[29,30].

The two species of *Lactobacilli* we predicted and tested in mice fed WD, enhanced systemic glucose tolerance, decreased adiposity, reduced several "bad lipids" in the liver, which could be all a consequence of improved hepatic mitochondrial health. This thought is supported, on the one hand, by clinical studies that have shown that reduction in hepatic fat in animals and humans results in recovery from T2D[37,51,52]. On the other hand, impairment of liver mitochondrial function has been long known as an important contributor to metabolic disease[33–35,53]. Furthermore, it has been shown that both palmitic and oleic acids (decreased by *Lactobacilli*) can damage liver mitochondria[54–56]. Conversely, enhancement of mitochondrial functioning stimulates beta-oxidation resulting in the reduction of damaging fatty acids[57,58].

The multi-omic network analysis in our study further supported the central role of hepatic mitochondrial health. Specifically, it pointed to several genes (Fig. 5a–c) involved in proper mitochondrial organization and mitochondrial autophagy (mitophagy) as the key players in relation to systemic glucose metabolism.

Investigations performed over the last decade have reported several mechanisms whereby microbiota can affect T2D including modulation of inflammation and immune mediators, gut hormones, mucosal permeability, insulin production among others[59]. Our present findings bring to the picture of host–microbiota interactions an intriguing link between mitochondria (regarded as

mammalian endosymbionts) and the symbiotic microorganisms in the gut. Interactions between mitochondria and microbiota is an emerging direction in microbiome research and have been implicated in Parkinson's disease[60], intestinal cell death by antibiotic-resistant microbiota[61] and longevity of *Caenorhabditis elegans*[62]. Metabolic health is synonymous with mitochondrial health where the ancestral mitochondrion-microbiome axis may play an important role[63].

Our investigation of serum metabolome pointed to several changes caused by *Lactobacilli*. Although the fact that *Lactobacilli* supplementation can alter certain bile acids levels might not be

surprising, a biological role of these alterations is uncertain. Furthermore, we were not able to follow-up the detected changes by targeted metabolomics in this work, which can be a subject of future studies. However, two metabolites, GSH and bilirubin, are known to play complementary antioxidant roles, which would improve mitochondrial respiration and other metabolic functions[64,65]. More recent reports demonstrated that deletion of biliverdin reductase A, which transforms biliverdin into bilirubin induced oxidative stress and lipid accumulation[66] and that bilirubin itself protects mitochondria via scavenging $O_2^{-}$[67]. GSH, however, uses somewhat different mechanisms of beneficial

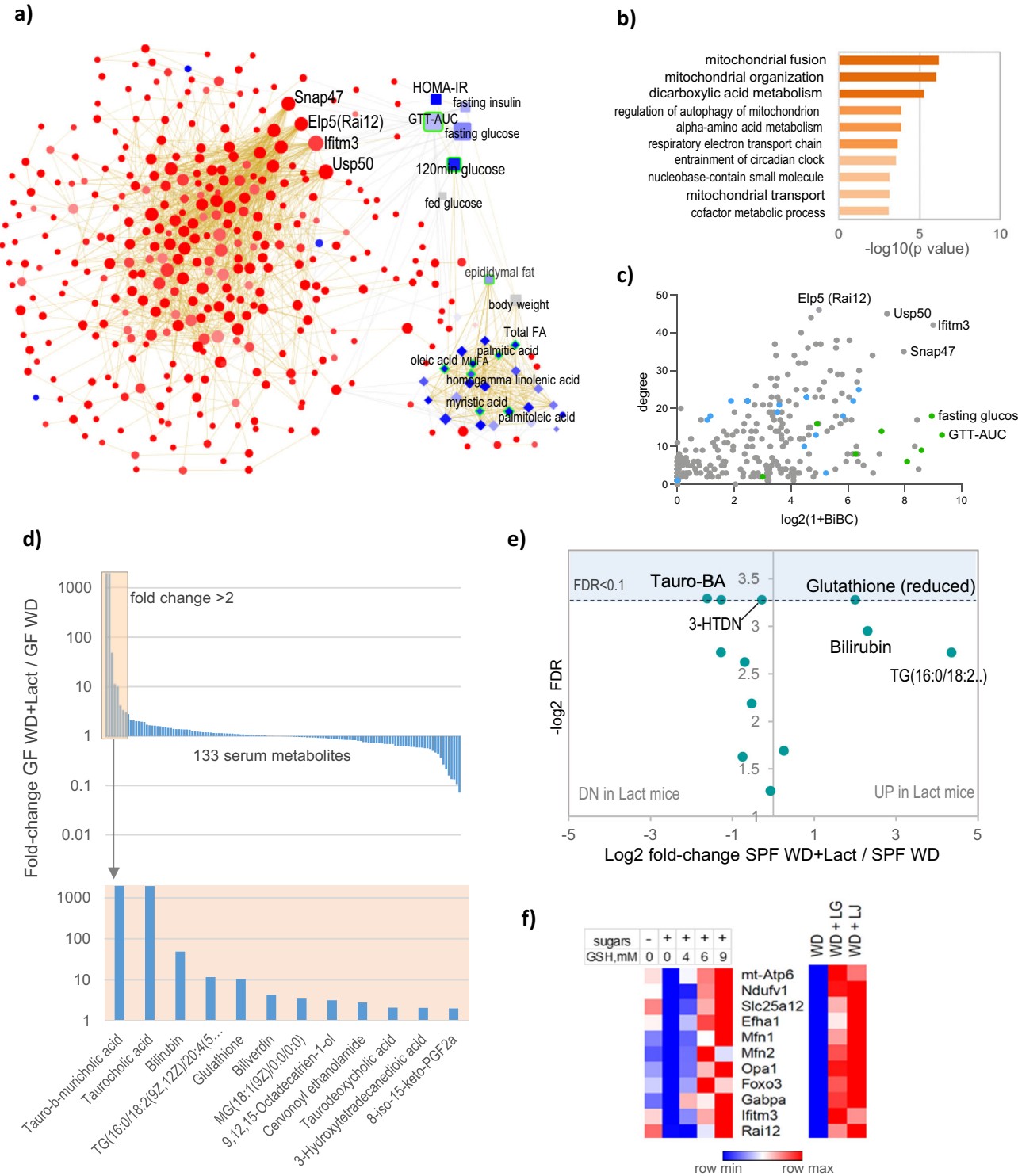

**Fig. 5 Multi-omic network analysis, metabolomics in mice supplemented with *Lactobacilli* and validation of glutathione in vitro. a** Multi-omic network integrating gene expression of genes significantly regulated in liver by *Lactobacilli* (circles), liver lipid profile (diamonds), and systemic metabolic parameters (squares) with red symbols indicating upregulated and blue are down in *Lactobacilli* supplemented mice. Green outline of nodes indicates significantly decreased lipid or phenotype; size of circle corresponds to the combined score of degree and bipartite betweenness centrality (BiBC) in the network. The orange and black edges indicate positive and negative correlations, respectively. Genes with top degree and BiBC are indicated. Source data are available at https://tinyurl.com/multi-omic-NW-Fig-5A. **b** Gene ontology biological functions over-represented in the genes of multi-omic network. **c** Scatterplot showing the degree and BiBC of all nodes in the multi-omic network with genes (gray), lipids (blue), phenotypes (green). **d** Fold-changes of 133 serum metabolites in germ-free (GF) mice fed western diet (WD) and colonized with *L. gasseri* for 2 weeks in comparison with GF mice on WD ($n = 2$ per group). TG, Triacylglycerol (16:0/18:2(9Z,12Z)/20:4(5Z,8Z,11Z,14Z)); MG, Monoacylglycerol; 8-iso-15-keto PGF2α, 8-iso-15-keto Prostaglandin F2α. Source data are provided in Supplementary Supplementary Data S14. **e** Changes in 12 metabolites identified in Fig. 5d in specific-pathogen mice (SPF) fed WD (data of serum pools of 4–6 mice in each pool per group), in five experiments of *Lactobacilli*-supplemented mice, mean fold change across five experiments and FDR (false discovery rate) is plotted. Source data are provided in Supplementary Supplementary Data S14c. **f** Left heatmap shows the geometric mean of normalized gene expression in AML-12 cells treated with either low sugar medium (glucose 17 mM), high sugar medium (glucose and fructose at 50 mM each) or high sugar medium supplemented with 4 mM, 6 mM, or 9 mM of reduced glutathione (GSH) ethyl ester (5–6 independent experiments). The right heatmap shows geometric mean of normalized gene expression from RNA-Seq in liver of western diet (WD) fed mice or WD-fed mice supplemented with either *L. gasseri* or *L. johnsonii* (red, high; blue, low relative gene expression). Source data are provided as a Source Data file.

effects on mitochondria. For example, it was shown to improve mitochondrial fusion[68]. Indeed, we found that both *Lactobacilli* in vivo and GSH in vitro increased expression of three main GTPases (Mfn1, Mfn2, Opa1) required for this process.

Unlike bilirubin, which is produced by hepatocytes, GSH origin is not limited to mammalian cells but it can also be produced by many bacteria. For example, some species of *Lactobacilli* are known to produce GSH, which they utilize to protect themselves from bile salts, reactive oxygen species and other types of cellular damages[69,70]. Therefore, it is plausible that our observation of increased levels of GSH is a result of simultaneous induction of its production by host cells[71] and by *Lactobacilli* itself. Although, further studies are warranted to identify the main source of GSH, it is highly plausible that this metabolite is one of the main mediators of *Lactobacilli* effect on liver mitochondria.

In agreement with our result, it was reported that another strain of *L. johnsonii* may improve hepatic mitochondria[72]. Interestingly, these mitochondrial effects may not be limited to the liver, as another species of Lactobacilli *L. paracasei* attenuated cardiac mitochondrial dysfunction in obese rats[73], and a different strain of *L. gasseri* increased resistance to mitochondrial dysfunction in aging *C. elegans*[74]. Notable, the two strains (*L. gasseri* and *L. johnsonii*) identified and tested in our study are also promising candidates for future testing in clinical settings of T2D as they would have minimal adverse effects on gut microbiota while improving glucose metabolism. Other strains of these two species of *Lactobacilli* have been tested in clinical trials for other diseases and in mouse models of diabetes[59,75] and thus might share critical mechanisms of effects on the mammalian host.

In conclusion, our study demonstrates that damaging effects of western diet on metabolism can be at least partially explained by decrease of beneficial microbes (e.g., *Lactobacilli*) and increase of pathobionts (e.g., *R. ilealis*) in gut microbiota, each of them acting via different host pathways. Furthermore, it revealed potential probiotic strains for treatment of T2D as well as critical insights into mechanisms of their action, offering an opportunity to develop targeted therapies of diabetes rather than attempting to restore "healthy" microbiota as a whole.

## Methods

**Mice and diets**. Seven weeks old, C57BL/6 male mice were purchased from Jackson Laboratories (Bar Harbor, Maine) and housed at Laboratory Animal Research Center (LARC) at the Oregon State University. After 1 week of acclimatization, mice were either switched to western diet (WD) D12451 containing 45% lard and 20% sucrose or to a matched normal diet D12450K (ND) produced by Research Diets (New Brunswick, NJ). Mice were on these diets for 8 weeks. Two independent experiments were performed with five mice per group in each experiment. Ethical approval for this work was obtained from the Oregon State

University Institutional Animal Care and Use Committee. The study complied with all relevant ethical regulations regarding the use research animals.

**Bacteria**. *L. gasseri* ATCC 33323 were purchased from American Type Culture Collection (ATCC, Manassas, VA). *L. johnsonii* NCC 533 were donated by Nestlé Culture Collection (Nestec Ltd., Nestlé Research Center Lausanne, P.O. Box 44, CH-1000 Lausanne 26). Both bacteria were grown anaerobically in MRS broth for 24 h at 37°C, colony-forming unit (CFU) was determined by serial dilutions, aliquoted in 15% glycerol stocks in cryovials and stored at −80°C. Before the gavage, the bacterial glycerol stocks were thawed, spun down, and resuspended in sterile phosphate-buffered saline (PBS). For *Romboutsia* experiment, active culture of *R. ilealis* DSM 25109 were purchased from the German Collection of Microorganisms DMSZ.

**Bacterial supplementation experiments**. For the microbial supplementation experiments, 8-week-old C57BL/6 mice were given either ND or WD or WD + *L. gasseri* (gavaged $1 \times 10^9$ CFU/mouse every other day) or WD + *L. johnsonii* (gavaged $1 \times 10^9$ CFU/mouse every other day) for 8 weeks. For the control, both ND and WD groups were gavaged with equal volume of PBS (0.2 ml per mouse). Two independent experiments were performed with 5–6 mice per group per experiment. For the treatment experiment, mice were fed ND or WD for 8 weeks when one group of WD mice was supplemented with *L. gasseri* (gavaged $1 \times 10^9$ CFU/mouse every other day). GTT was performed at 8 weeks on WD and 4, 9, and 12 weeks on WD + *L. gasseri* ($n = 5$ per group). For *R. ilealis* supplementation experiment, after 1 week of acclimatization, all mice were switched to ND and were either given PBS or $1 \times 10^9$ CFU of *R. ilealis* every other day for 4 weeks ($n = 5$). Metabolic measurements were done as described below except for *R. ilealis* experiment 1 mg/kg glucose was injected for IPGTT.

For gnotobiotic mouse experiment, germ-free mice on western diet were colonized with $1 \times 10^9$ CFU *L. gasseri* on Day 0, Day 2, Day 4, and Day 12 and killed on D14 ($n = 2$).

**Intraperitoneal glucose tolerance test (IPGTT)**. Mice were fasted for 6 h during the light phase with free access to water. A concentration of 2 mg/kg glucose (Sigma-Aldrich) was injected intraperitoneally. Blood glucose was measured at 0 min (immediately before glucose injection), 15, 30, 60, and 120 mins with a Freestyle Lite glucometer (Abbot Diabetes Care).

**Fasting insulin and fasting glucose**. Mice were fasted for 6 h with free access to water. Fasting blood was collected either via submandibular bleed or from the tail vein. Insulin and glucose levels in fasting plasma or serum was measured with Mouse Insulin ELISA Kit (Crystal Chem) and Glucose Colorometric Assay Kit (Cayman Chemical), respectively, according to manufacturer's protocol. HOMA-IR and HOMA-B were calculated according to Eqs. (1) and (2), respectively:

$$HOMA - IR = \frac{Glucose\,(mg/dL) \times Insulin(\mu U/mL)}{405} \quad (1)$$

$$HOMA - B = \frac{360 \times Insulin\left(\mu \frac{U}{mL}\right)}{Glucose\left(\frac{mg}{dL}\right) - 63}\% \quad (2)$$

The heatmap of results of systemic measurements was created using Morpheus (https://software.broadinstitute.org/morpheus/).

**Hepatic fatty acids and cholesterol**. Hepatic fatty acids were quantified using established protocols[76]. In brief, total lipid was extracted from liver in

chloroform–methanol (2:1) containing 1 mM butylated hydroxytoluene. 7-Nonadecenoic acid (C19:1) was added as a recovery standard. Total protein was measured after the initial homogenization step by bicinchoninic acid assay (Bio-Rad, Hercules, CA). Fatty acids in the extracts were saponified in 80% methanol containing 0.4 M KOH. Afterward, saponified fatty acids were converted to fatty acid methyl esters in methanol containing 1% of 24 M $H_2SO_4$ and then quantified by gas chromatography.

Hepatic total cholesterol in liver lipid extracts and in serum was measured using Amplex™ Red Cholesterol Assay Kit (Thermo Fisher Scientific) according to manufacturer's protocol.

**RNA preparation and gene expression analysis.** RNA was extracted using an OMNI Bead Ruptor and 2.8 mm ceramic beads (OMNI International) in RLT buffer followed by Qiashredder and RNeasy kit using Qiacube (Qiagen) automated extraction according to manufacturer's specifications. Total RNA was quantified using Quant-iT RNA Assay Kit (Thermo Fisher Scientific). Complementary DNA was prepared using qScript reverse transcription kit (Quantabio) and qPCR was performed using Perfecta SYBR mix (Quantabio) and StepOne Plus Real Time PCR system and software (Applied Biosystems). RNA libraries were prepared with QuantSeq 3′mRNA-Seq Library Prep Kit (Lexogen) and sequenced using Illumina NextSeq. Sequences were processed to remove adapter, polyA and low-quality bases by BBTools (https://jgi.doe.gov/data-and-tools/bbtools/) using bbduk parameters of $k = 13$, ktrim = r, forcetrimleft = 12, useshortkmers = t, mink = 5, qtrim = r, trimq = 15, minlength = 20.

Reads were aligned to mouse genome and transcriptome (ENSEMBL NCBIM37) using Tophat (v2.1.1) [77] with default parameters. Number of reads per million for mouse genes were counted using HTSeq (v 0.6.0)[78] and quantile normalized. BRB-ArrayTools was used to identify genes differentially expressed in the liver and ileum when supplemented with or without the *Lactobacillus* candidates. Pathway enrichment was performed using Metascape[79].

**DNA extraction and 16 S rRNA gene libraries preparation.** For microbial measurements, stool pellets were collected at T1 (4 weeks of diet) and stool pellets and terminal ileum contents were collected at T2 (8 weeks). To get microbial DNA, frozen fecal pellets, and ileum with content were resuspended in 1.4 ml ASL buffer (Qiagen) and homogenized with 2.8 mm ceramic beads followed by 0.5 mm glass beads using an OMNI Bead Ruptor (OMNI International). DNA was extracted from the entire resulting suspension using QiaAmp mini stool kit (Qiagen) according to manufacturer's protocol. DNA was quantified using Qubit broad range DNA assay (Life Technologies). The V4 region of 16 s rRNA gene was amplified using universal primers (515 f and 806r) as in ref. [16]. Individual samples were barcoded, pooled to construct the sequencing library, and then sequenced using an Illumina Miseq (Illumina, San Diego, CA) to generate pair-ended 250 bp reads.

**16 S rRNA gene sequencing data analysis.** The samples were demultiplexed and forward-end fastq files were analyzed using QIIME v. 1.9.1[80]. The default quality filter parameters from QIIME's *split_libraries_fastq.py* were applied to retain high-quality reads (Phred quality score ≥ 20 and minimum read length = 75% of 250 nucleotides). A closed reference OTU picking with 97% sequence similarity was performed using UCLUST[81] and Greengenes reference database v13.8[82,83] to cluster 16 S rRNA gene sequence reads into OTUs and assign taxonomy. The reference sequence of candidate OTUs from the Greengenes database was used to obtain species level taxonomic assignment using Megablast[84] (top hit using default parameters). A threshold of 99% cumulative abundance across all samples in an experiment was used to retain abundant microbes, thus removing OTUs with ~<0.01% abundance across all samples in that experiment. The read counts were normalized using cumulative sum scaling[85], accounted for DNA quantity, followed by quantile normalization. The principal component analysis for the 16 S sequencing data was created using Clustvis[86], GraphPad Prism software (version 7), R packages seqtime version 0.1.1, igraph version 1.2.5.

**Network analyses**

*TK Network reconstruction and prediction of causal microbes.* Spearman rank correlations were calculated between all pairs of microbes (OTUs) and metabolic parameters (phenotypes) in each group of both experiments. A combined Fisher's $p$ value was calculated for each pair from the correlation $p$ values from each experiment. A FDR was calculated on the combined $p$ values separately for the following correlations: (i) within metabolic parameters, (ii) within OTUs, and (iii) between OTUs and metabolic parameters. We retained edges that satisfied the following criteria: the sign of correlation coefficients in the two experiments consistent in stool of WD-fed mice at 4 weeks ($n = 35$ per expt.), individual $p$ value of correlation within each experiment is <30%, combined Fisher's $p$ value of all experiments <5% and FDR cutoff of 10% for within edges (i and ii). Finally, the TK network was generated[20,61,87–89] by adding microbe-phenotype edges where the microbe showed significant change in (WD vs ND) abundance in ileum at 8 weeks, edges showed consistent sign of per group Spearman correlation coefficient between the two experiments of three WD-fed groups (WD-stool 4 weeks, WD-stool 8 weeks, and WD-ileum 8 weeks), and satisfied principles of causality[90] (i.e., had concordance between fold change in WD vs. ND comparison and

correlation sign between the two partners) in all three WD-fed groups. The network was visualized in Cytoscape.

**Identification of keystone microbes.** Generation of training data were accomplished as follows: 100 instances of 542 generalized Lotka-Volterra models were run to steady state and steady state species abundances were considered individual samples. Those individual samples consisted of 10–100 species drawn from a model-specific species pool. The size of the species pool was determined by defining similarity in species composition between samples (between 0.4 and 0.95). The individual models further varied in the following parameters: connectivity of the species interaction matrix (between 0.005 and 0.7), negative edge percentage of the species interaction matrix (0–100%), species-specific growth rates (between 0 and 1) and carrying capacities (between 0 and 100), as well as the topography of the species interaction matrix (interactions sampled from a uniform distribution or assigned according to the Klemm-Eguíluz model[91]. The R-package seqtime was used to generate the species interaction matrices[92].

Subsequently, each species included in a model was in turn removed from the community and a Canberra distance between original and sub-sampled community was calculated. In all, 1000 iterations of this procedure were performed per species and the average Canberra distance induced by a species' absence was considered its keystoneness score.

For Model training, the data were split into training set and test set. The training set was used to train a linear model to predict keystoneness based on mean relative abundance and the following node parameters computed from a spearman correlation network: sum of absolute correlation strength, node degree, relative closeness centrality, betweenness centrality, and eccentricity. With the exception of absolute correlation strength, the network parameters were calculated within the R-package igraph (http://igraph.org). This model was then used to predict keystoneness on the test set. A linear model between real and predicted keystoneness in the test set gave an adjusted $R^2$ of 0.4219, with a $p$ value <2.2e-16.

The trained linear model was subsequently applied to the OTU abundance data and the previously computed correlation network to predict keystoneness scores for each OTU. At last, keystoneness scores were scaled between 0 and 1 to remove negative values occurring as an artifact of the linear model.

**Multi-omic network analysis.** Spearman rank correlations were calculated between all pairs of genes, lipids, and phenotypes. The phenotypic subnetwork was obtained from the TK network. For gene subnetwork, correlation was calculated by pooling samples supplemented with the same Lactobacilli from both experiments. Edges were retained if they satisfy the following criteria: the sign of correlation coefficients in the two Lactobacilli groups should be consistent, individual $p$ value of correlation is <30%, combined Fisher's $p$ value over two Lactobacilli groups <5%, FDR cutoff of 5%, and satisfying principles of causality (i.e., satisfied fold change relationship between the two partners in the Lactobacilli vs. WD comparison). For the lipid subnetwork, correlations were calculated per experiment in the WD groups of the three datasets (two WD vs ND experiments, and a Lactobacilli supplementation experiment). Edges were retained if the sign of correlation coefficients was consistent, Fisher's $p$ value <5%, FDR cutoff of 10%, and satisfied principles of causality.

For between-omics edges, correlations were calculated per experiment in the WD groups of three data sets and a voting strategy was used for meta-analysis. Pairs were shortlisted if they had the same sign of correlation and $p$ values <10% in at least two data sets. If the $p$ value in the third data set was over the threshold, the pair was retained but the third data set was removed during calculation of Fisher $p$ value. The pair was kept if the p-value in the third data set was under the threshold and the sign of correlation was same in all three data sets, else the pair was entirely removed. Edges with FDR < 10% and satisfying principles of causality were added to the network.

**Computational analysis using human datasets.** Sequence read files of 1046 humans[25] were downloaded from European Bioinformatics Institute (https://www.ebi.ac.uk/), quality filtered, and trimmed with ea-utils using default settings except the base removal quality threshold was set at <20. Cleaned sequence reads were binned into Greengenes (v13_8) 97% identity OTUs using the QIIME 1.9 closed reference OTU picking workflow (pick_closed_reference_otus.py). Spearman correlations between BMI and microbial abundance of exact candidate OTU (or the sum of OTUs assigned to the bacterial species) were calculated in obese humans. To avoid bias from outlier samples, a sample was considered only if had > 10 reads per million for *Lactobacillus* OTUs and >100 reads per million for *Romboutsia* OTUs.

**Transmission electron microscopy (TEM).** Frozen liver samples were prepared and fixed in 1.5% paraformaldehyde and incubated at 4 °C overnight[93], after which fixed tissues were processed usinf a protocol based on ref. [94]. Specifically, the vibratome sectioned fixed tissues (~1 mm³) were postfixed in solution containing 2% osmium tetroxide and 1.5% potassium ferrocyanide for 30 min at room temperature in dark. It was followed by staining with 0.2% tannic acid in water for 10 min, fixing in 1% osmium tetroxide for 30 min and staining in 1% thiocarbohydrazide in water for 20 min at room temperature. The samples were then incubated with 1% osmium tetroxide for 30 min at room temperature. Then the

samples were incubated with 0.5% uranylacetate in 25% methanol overnight at 4oC, which was followed by incubation in Walton's lead aspartate for 30 min at 60 C. Then samples were dehydrated with graded series of ethanol, infiltrated with ethanol/epon mixture (1:1) for 1 h at room temperature and 1:2 for 1 h at room temperature. Ultramicrotome was done using a RMC PowerTome PC. Microscopy was done with a Helios 650 NanoLab (ThermoFisher). Scanning transmission electron microscopy mode was used for imaging. In all, 10–12 images were taken per sample. The images were imported into FIJI (i.e. ImageJ) software (version 2.0.0-rc-69/1.52i). Each mitochondrion in the images was outlined and different attributes were measured using default "measure" option in the software.

In order to identify image parameters that discriminate between healthy and damaged mitochondria, we used images representative of all analyzed groups. In each image, a pair of damaged (bright, lucent) and healthy mitochondria (dark, dense) were identified according to images in EM atlas (http://www.drjastrow.de/WAI/EM/EMAtlas.html). Next, we extracted quantitative data for 17 different image parameters (See Supplementary Data 11) and analyzed which of those differed between the two types of mitochondria. The selection has been performed "blindly" (i.e., the image analyst was unaware of treatment identity of samples). Among parameters that significantly differed between two types of mitochondria we chose less interdependent ones to compare different treatment groups. To establish whether the structure of mitochondria differs between groups supplemented or not with probiotic bacteria we analyzed the above selected image parameters in 119 TEM images from liver samples of nine mice totalizing 4709 mitochondria.

**Un-targeted metabolomics.** Serum samples used for metabolomics included the following: germ-free mice fed WD for 2 weeks ($n = 2$), monocolonized for 2 weeks with *L. gasseri* fed WD ($n = 2$); SPF mice supplemented or not with either *L. gasseri* or *L. johnsonii* ($n = 4$–6 per group) and fed WD for 8 weeks in two experiments shown in Fig. 3; SPF mice first fed WD for 8 weeks, then supplemented (or not) with *L. gasseri* for additional 12 weeks along with WD ($n = 5$ per group). For technical reasons, metabolomics was performed in pooled sera of each group of mice, which were run in a randomized manner as one batch.

An aliquot of 30 µl of pooled serum was processed following a protocol adapted from a published study[95]. In brief, metabolites were extracted with four volumes of cold methanol/acetonitrile (1:1, v/v). To precipitate proteins, the samples were incubated for 1 h at −20 °C. After the samples were centrifuged at 4 °C for 15 min at 15,871 × g (13,000 rpm), the supernatant was collected and evaporated to dryness in a vacuum concentrator. The dry extracts were reconstituted in 90 µL of acetonitrile/H2O (1:1, v/v) containing 10 ng/mL CUDA (12-(((cyclohexylamino)carbonyl) amino)-dodecanoic acid). This standard was used as a control to monitor platform stability along the fully randomized batch analysis, and to account for possible injection variabilities. A quality control (QC) pooled sample was prepared by combining, in a single vial, 10 µL of each sample. Pooled QC sample provided a 'mean' profile representing all analytes encountered during the analysis. To the QC sample a methanol solution containing verapamil and verapamil-D3 (Cayman Chemical, Ann Arbor, MI) was added at a final concentration of 0.1 ppm each. The ratio of their monoisotopic peaks was used to monitor quantification stability along the fully randomized batch analysis. The supernatant was then analyzed via LC-MS/MS (liquid chromatography with tandem mass spectrometry).

High-resolution mass spectrometry was performed using an Agilent 6545 Q-ToF downstream of an Agilent 1260 Infinity high-performance liquid chromatography system consisting of a degasser, quaternary pump, autosampler (maintained at 4 °C) and column heater (maintained at 30 °C). The Q-ToF machine was operated using MassHunter software and an analysis in positive and negative ionization mode was performed for each sample. Separation was achieved using an InfinityLab Poroshell EC-C18 column (100 × 3.0 mm, 2.7 µm, Agilent) at a flow rate of 0.4 mL/min. Line A was water with 0.1% (v/v) formic acid and line B was methanol with 0.1% (v/v) formic acid, adapted from a previously described protocol[96]. The column was pre-equilibrated with 1% B. After injection (3 µL of the sample) this composition was held for 1 min and then changed to 30% B over the next 10 min using a linear gradient. The composition was then changed to 100% B over the next 14 min and then held at 100% B for 5 min. The mobile phase was then adjusted back to 1% B over two minutes and the column was re-equilibrated for 6 min prior to the next injection. The Agilent Q-ToF mass spectrometer was equipped with an Agilent JetSpray source operated with the following parameters: Auto MS/MS mode, Gas Temp, 325 °C; Drying gas, 10 L/min; Nebulizer, 20 psi; Sheath gas temp, 375 °C; Sheath gas flow, 12 L/min; Capillary Voltage (VCap), 4000 V; Nozzle voltage (Expt), 600 V; Fragmentor, 175 V; Skimmer, 65 V; Oct 1 RF Vpp, 750 V; Mass range, 100-3000 m/z; Acquisition rate, 10 spectra/s; Time, 100 ms/spectrum. The MS/MS spectra (mass range, 50–3000 m/z; acquisition rate, 10 spectra/s; time, 100 ms/spectrum) were obtained by isolating the precursor ion with a medium isolation width (~4 m/z) summing spectra generated with collision energies of 15, 30, and 40 V. Blanks and QC samples were run before and after every four serum samples and the column was re-equilibrated. Based on the reproducibility of our QC and on the intensity of the CUDA, we can assume that the instrument was stable during the full randomized batch, and that intensity differences are due to biological differences and not to technical variation.

**LC-MS/MS data processing.** Raw data were imported into Progenesis QI software (Version 2.3, Nonlinear Dynamics, Waters) in order to perform data normalization, feature detection, peak alignment, and peak integration[97–99]. Metabolites were confirmed by MS, MS/MS fragmentation, and isotopic distribution using Metlin (Version 1.0.6499.51447, https://metlin.scripps.edu) and the Human Metabolome (Version March 2020, https://hmdb.ca) databases as the reference[100]. The data acquired in both, electrospray ionization (ESI) negative and positive modes, which resulted in ESI+ in 7100 features with just MS information, 2461 features with both MS and MS/MS information; serum ESI− gave 2141 features with just MS information and 1204 features with both MS and MS/MS information. Thus, a total of 3665 features with both MS and MS/MS information was obtained. Next, a metabolite was sieved out when a match with a difference between observed and theoretical mass was <10 ppm and the molecular formula of matched metabolites further identified by the isotopic distribution measurement. By doing so, the number of annotated compounds with a known identification was reduced to 133 metabolites, which had match score >35 (range 36.1–57.8), and isotope similarity between 67.8 and 99.1%). We chose to increase the confidence of our annotations, rather than increase the number of annotated compounds with a lower level of confidence. Zero values were assigned minimal values calculated as three STDEV of technical variation subtracted from the minimal measured level of a given metabolite in this study. Technical variation was defined by using CUDA and corresponded to STDEV of 0.135 and mean of 1.02. The level of metabolite identification was 2 for all compounds based on Sumner et al.[101]: level two refers to putatively annotated compounds (e.g., without chemical reference standards, based upon physicochemical properties and/or spectral similarity with public/commercial spectral libraries).

**Cell culture.** AML-12(ATCC CRL-2254) cells were grown in complete growth medium (DMEM:12 Medium (ATCC 30-2006) supplemented with 10% fetal bovine serum (FBS), 10 µg/ml insulin, 5.5 µg/ml transferrin, 5 ng/ml selenium, 40 ng/ml dexamethasone, and 1% penicillin/streptomycin) at 37 °C in 5%CO2. After obtaining 80–85% confluency, 20,000 cells per well were seeded in complete growth medium in 96 well plate for 24 h. After 24 h of incubation, the medium was replaced either with low glucose medium (5.5 mM Glucose, 10% FBS, low sugar group) or mixture of 100 mM Glucose and Fructose (1:1 ratio, with 10% FBS, high sugar group) alone or mixed with 4, 6, or 9 mM reduced GSH ethyl ester (GSH, Sigma-Aldrich). After 6 h of treatment, culture medium was removed, cells were lysed in RLT buffer (Qiagen) and RNA was extracted using RNeasy Mini kit (Qiagen). Total RNA was quantified using Quant-iT RNA Assay Kit (Thermo Fisher Scientific). Complementary DNA was prepared using qScript reverse transcription kit (Quantabio) and qPCR was performed using Perfecta SYBR mix (Quantabio) and StepOne Plus Real Time PCR system and software (Applied Biosystems). Polymerase (Polr2c) gene was used as the control gene. Primers used for qPCR are listed in the supplementary Supplementary Data 15b. Total six experiments were performed. The gene expression was normalized using the control group per experiment and per gene across the experiments, followed by log2 transformation. Control and treatment groups were compared using paired, one-sided parametric t test.

**Statistics and reproducibility.** Overall, the data were log transformed, checked for normality and an appropriate test was performed accordingly (i.e., parametric tests as default and non-parametric tests when distribution did not fulfill normality criteria), followed by Benjamini–Hochberg false discovery rate correction. A two-sided test was used when there was no prior hypothesis of the expected direction of change; otherwise, one-sided test was used. For initial experiments, to capture the strongest and consistent signals across independent experiments (e.g., WD vs ND), non-parametric tests were used, and the meta-analysis was performed over experiments using Fisher's meta-analysis test. To achieve statistical power in the Lactobacilli supplementation experiments, the samples were normalized within each experiment to the mean of control group and analyzed together using parametric tests for host-derived variables. Meta-analysis was performed over the microbiome data. Gene enrichment analysis using Metascape software[79] that implements hypergeometric test. For metabolomics analysis, results of five lactobacilli supplementation from three experiments were normalized over corresponding controls with no probiotic supplementation. Log2 transformed ratios (lacto/control) for each metabolite were compared for deviation from 0 using parametric test. In experiments with interrelated data from two groups (e.g., AML-12 in vitro experiment) we used paired test. Outliers (1%) were identified using ROUT method of GraphPad Prism 8.4.1 and removed (used only once in the whole study, one value was removed for one concentration of GSH treatment). Actual tests, cutoffs applied are mentioned in each figure caption, exact p values are available in supplementary data and source data files.

**Reporting summary.** Further information on research design is available in the Nature Research Reporting Summary linked to this article.

## Data availability

Data were submitted to NCBI SRA under submission PRJNA558801 for 16 S rRNA, to GEO under GSE136033, and to Metabolomics Workbench under ST001436. TK network

access: https://tinyurl.com/TK-NW-Fig-1C. Multi-omic network access: https://tinyurl.com/multi-omic-NW-Fig-5A. Source data are provided with this paper.

## Code availability

Custom codes available at https://github.com/richrr/TransNetDemo and https://github.com/fbauchinger/keystone_species_model.

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

## Acknowledgements
We thank Teresa Sawyer from the OSU Electron microscopy core for excellent service, Drs. Jodi Nunnari, Chrissa Kioussi, Matthew Robinson for advice regarding mitochondria, Claudia S. Maier, Director of the OSU Mass Spectrometry Center for providing access to diverse software tools and advice regarding metabolomics, Laboratory Animal Resource Center (LARC) and Center of Genome Research Biocomputing (CGRB) at OSU for technical support.

## Author contributions
Original idea and overall study design: A.M., NS. Design of individual experiments: R.R.R., M.G., Z.L., M.G.J., R.G., B.P., D.B.J., G.T., AD, A.M., N.S. Data generation: M.G., Z.L., M.G.J., R.G., H.Y., J.W.P., C.G., S.V.P., K.D.W., B.F., B.P., D.B.J., A.D. Data analysis: R.R.R., M.G., Z.L., M.G.J., F.B., T.J.S., C.G., B.P., D.B.J., G.T., D.B., A.D., A.M., N.S. Drafting manuscript: R.R.R., M.G., Z.L., M.G.J., F.B., B.P., A.M., N.S. Editing manuscript: T.J.S., B.P., D.B.J., G.T., D.B, A.M., N.S. Supervision of specific set of experiments and/or series of data analyses: T.J.S., B.P., D.B.J., G.T., D.B., A.M., N.S. Overall study leadership: A.M., N.S.

## Funding
NIH R01 DK103761 (NS), DK112360 (DBJ), European Research Council starting grant FunKeyGut 741623 (D. Berry and F.B.).

## Competing interests
The authors declare no competing interests.
