## [Peer Review File · Nature Communications]

Reviewers' Comments:

Reviewer #1:

Remarks to the Author:

I read with interest this paper which addresses an interesting question.

The authors first identified 4 OTUs (identified one *R. gnavus*, *R. ilealis*, and 2 lactobacilli) that may improve or worsen the development of type 2 diabetes upon western diet. In a second step, the authors move on to validate their findings using correlations between abundance of these bacteria and BMI in a human cohort. The author's claim of such validation is not supported by the data (Fig 2, only 2 correlations on the 6 tested are significant and one bacteria – *R. gnavus* – was not tested, for an unknown reason). In a third step, the authors tested 3 bacterial strains from the same species as the OTUs identified and claim that these bacteria worsened glucose homeostasis. In a fourth and last step, the authors looked for a mechanism for both lactobacilli while *R. ilealis* was left behind. They show that lactobacilli supplementation changes mitochondria size and counteracts fatty acid profile.

The data provided do not support the title as none of the experiments demonstrate that the ability of lactobacilli to change hepatic mitochondria size is key for the improvement of diabetes in mice. The fact that lactobacilli can improve metabolic disorders associated with WD and mitochondrial function has been reported by other teams.

Why the primary pathological outcome seems to change from one step to another (glucose homeostasis versus BMI) is not clear to me.

There was no explanations on how the strains were selected while the authors acknowledge that several strains from the identified species have been previously tested.

I struggled to understand the statistics behind most of the figures. No statistical subsection was present in the material and method section. What was the level of significance? Why is it different from one set of analysis to another?

A few specific examples:

- For the gut microbiota analyses, in the legend of figure 1e, on which type of data was the Fisher test computed?
- In Fig 3a, no statistics are provided for the glucose tolerance curves. What was the test used to compare the 2 supplemented groups to the unsupplemented group?
- The authors claim that "Conversely, mice supplemented with *R. ilealis* showed impaired glucose tolerance (AUC and 15 mins. glucose levels)". However, no significant difference is shown in the related graph (Fig 3a, last graph).
- Fig 4e-h, what was the test used? Are the data displayed with SD or SEM? In Fig 4f, the difference between the WD group and the lactobacilli groups is quite low, is that biologically relevant?

The authors state that "Recently, there were several reports concerning the effects of pathogens and probiotics on the gut microbiota and a possibility that their effect on the host is mediated by those alterations (27,28). Therefore, we sequenced 16S rRNA gene in ileum and fecal samples from mice supplemented with three candidate bacteria. In contrast to other studies, very few changes were observed in the ileal and stool microbiota composition due to supplementation by these microbes".

Refs 27 and 28 are quite outdated and several reports have now repeatedly shown that probiotics/ bacteria supplementation, in their large majority, do not affect gut microbiota taxonomic composition (Reviewed in Kristensen, et al, *Genome Medicine* 2016; also shown in Simon et al *Diabetes Care* 2015; Maldonado-Gomez et al, *Cell Host & Microbe* 2017; <https://isappscience.org/for-scientists/resources/probiotics/>)

Reviewer #2:

Remarks to the Author:

The major claim of this manuscript is that microbes interact (positively and negatively) with liver mitochondria in a way that modifies the outcome of T2D-related tests such as glucose intolerance. As per the title, this may mean that Lactobacilli are capable of interacting with liver mitochondria in order to attenuate diabetes induced by a western diet in mice.

Overall, I find this to be interesting proof-of-principle for the effect of microbes from the gut microbiome on the severity of T2D. Given that some of the results appear to have small effect sizes (but have statistical significance), the biggest question in my mind is whether pro or prebiotics will be useful in humans, as the authors imply. That said, I do think that this manuscript will change the thinking in the field and I also think the authors have appropriately limited their conclusions. There is still not a strong consensus or awareness of the links between the microbiome and mitochondria - likely co-evolutionary partners - and the work in this manuscript will add to an important body of literature. More specifically in this work, the authors show that changes in gene expression are affected by alterations in the gut microbiota. This is a hallmark in the sense that this is a study design the probes cause and effect (unlike past studies that looked at genetically modified mouse mitochondria or other heavier handed tweaks). I think the weaker effect size is in that sense realistic.

The largest change that would make this more convincing would be to add a mechanistic element to the analysis. For example, the mitochondria gene expression correlations in Fig 4b are significant, but somewhat unexplained. This leaves me a little confused as to what to make of these seemingly random mitochondrial functional categories. In addition, the transkingdom network is hard to read. Given the importance and prominence of the transkingdom network in the title, abstract, and text, I would have expected to be able to see the network and understand the connections that the authors draw between OTUs and who is an influencer, improver, etc.,. If there was a better way to make that clear, this would improve the impact of the manuscript enormously. Similarly, it seems like really the understandable element is how the nodes are scored for keystone-ness, but that is also confusing in places. For instance, in Fig 1d, it appears that the species that are ultimately tested are not the 4 most keystone species. If the logic is that keystone-ness led to the future tests, why were some of the more keystone species skipped?

Overall, I think this manuscript is appropriate for publication and could be strengthened by addressing some of the above points.

Reviewer #3:

Remarks to the Author:

The current manuscript investigates the effects of two candidate microbes on glucose homeostasis and links these effects to liver mitochondrial function. The studies are interesting and largely novel.

Major Comments:

1. There is no data linking the changes in glucose and mitochondria to alterations in liver triglycerides or liver histology.
2. There is no mechanism to explain why glucose tolerance is altered.
3. What is the mechanism for the reduction in adiposity despite no change in body mass?

Dear reviewers,

Thanks for your questions and suggestions. It appears that working on them revealed some novel, not necessarily expected details of the mechanisms of beneficial effects of the two species of Lactobacilli we worked on.

To answer to your questions we have performed additional network analysis integrating liver gene expression with systemic parameters and liver fatty acid data that pointed to the interaction between mitochondrial gene expression and glucose metabolism.

More importantly, we did two new experiments by supplementing Lactobacilli to germfree mice fed western diet and to regular (SPF) mice in which the disease had been established. Next, we have evaluated serum metabolomes of all supplementation experiments (in pools) revealing metabolites potentially mediating effects of probiotics on mitochondrial liver function, one of which we have validated *in vitro*.

Finally, these new studies resulted in the new Figure 5 and several supplemental figures and tables. Please find our point by point answers below. The new text in the manuscript is marked.

Reviewer #1 (Remarks to the Author):

I read with interest this paper which addresses an interesting question. The authors first identified 4 OTUs (identified one *R. gnavus*, *R. ilealis*, and 2 lactobacilli) that may improve or worsen the development of type 2 diabetes upon western diet. In a second step, the authors move on to validate their findings using correlations between abundance of these bacteria and BMI in a human cohort. The author's claim of such validation is not supported by the data (Fig 2, only 2 correlations on the 6 tested are significant and one bacteria – *R. gnavus* – was not tested, for an unknown reason).

We thank the reviewer for bringing up this point. We mistakenly used word 'validation' in respect to the analysis of human data. Our goal of using human data was to check the relevance of our top predicted taxa for human diabetes/obesity. The validation actually comes from the next step, which is the experiments of supplementing bacteria to mice. We re-phrased corresponding sentences and replaced 'computational validation' for 'computational verification'.

Regarding *R. gnavus*, since it was predicted as OTUs with moderate/minor negative effect on metabolism, we haven't originally included it in analysis of human data. However, a new analysis showed a trend towards weak positive correlation with BMI which agrees with inference in mice (i.e. low but negative effect $r=0.06$ $p<0.09$). Because of this lower ranking (both in mouse and human data) we haven't tested *R. gnavus* but we now included this graph in supporting figures. Interestingly, though, this microbe was previously reported in association with obesity (refs. 22, 23, PMID 24013136).

- In a third step, the authors tested 3 bacterial strains from the same species as the OTUs identified and claim that these bacteria worsened glucose homeostasis.

The reviewer is not entirely correct here. We tested 2 strains of the genus *Lactobacillus* that were predicted by our analysis and one of a different genus, which is *Romboutsia*.

- In a fourth and last step, the authors looked for a mechanism for both lactobacilli while *R. ilealis* was left behind. They show that lactobacilli supplementation changes mitochondria size and counteracts fatty acid profile.

The data provided do not support the title as none of the experiments demonstrate that the ability of lactobacilli to change hepatic mitochondria size is key for the improvement of diabetes

in mice. The fact that lactobacilli can improve metabolic disorders associated with WD and mitochondrial function has been reported by other teams.

A. We have two arguments supporting our suggestion that restoration of liver mitochondrial health contributes to improvement systemic glucose metabolism.

First, and probably the most important, extensive literature shows that damage to mitochondria by the excess of fat/sugar is one of the critical processes in the pathogenesis of type 2 diabetes (e.g. PMID: 30168804, 19838201, 26476631, 28300161). One paper (Eur J Nutr. 2016 Feb;55(1):1-6), after providing literature review concluded that “dysfunctional mitochondria play an important role in the development of insulin resistance and ectopic fat storage in the liver, thus supporting the idea that mitochondrial dysfunction is closely related to the development of obesity, type 2 diabetes mellitus”

Importantly, boosting hepatic mitochondrial functions was shown to improve diabetes phenotype (e.g. PMID: 31130874, 28223285, 27284107).

Second, our new unbiased network analysis showed that several genes involved in mitochondrial organization/repair (see Fig. 5) are connected to the main glucose metabolism parameters and not to other phenotypes such as body weight and fat. Although, only perturbation experiments (such as KOs) would provide complete evidence of their action these results are highly suggestive of liver mitochondria at least partial control over systemic glucose metabolism.

Thus, putting together our results and the existing literature support that Lactobacilli beneficial effect in WD induced T2D model is at least partially occurs through restoration of liver mitochondria.

B. Regarding lactobacilli strains improving hepatic mitochondria, to the best of our knowledge, there is only one report (Applied Microbiology and Biotechnology 98, 6817-6829, doi:10.1007/s00253-014-5752-1 (2014)), which didn't not study diabetes but rather fatty liver disease and did not comprehensively evaluate mitochondrial processes as we did. Furthermore, a simultaneous induction of glutathione and bilirubin by lactobacilli is certainly novel and potentially very important effect in light of complementary antioxidant effects produced by these two metabolites (Proc Natl Acad Sci U S A. 2009 Mar 31; 106(13): 5171–5176.) We appreciate if the reviewer can provide any additional references.

➤ Why the primary pathological outcome seems to change from one step to another (glucose homeostasis versus BMI) is not clear to me.

We used BMI for human data analysis since this was the only available parameter in the analyzed human dataset. In mouse studies adiposity is the corresponding parameter, including in the original study we used (ref. 26).

➤ There was no explanations on how the strains were selected while the authors acknowledge that several strains from the identified species have been previously tested.

To select the strains, we used BLAST to align sequences of OTUs against the NCBI microbial nucleotides. The strains with the highest similarity (> 90%) were selected. We provide in a new supplementary table the information about each OTU's similarity to the selected strain.

- I struggled to understand the statistics behind most of the figures. No statistical subsection was present in the material and method section. What was the level of significance? Why is it different from one set of analysis to another?
A few specific examples:
 - For the gut microbiota analyses, in the legend of figure 1e, on which type of data was the Fisher test computed?
 - In Fig 3a, no statistics are provided for the glucose tolerance curves. What was the test used to compare the 2 supplemented groups to the unsupplemented group?
 - The authors claim that “Conversely, mice supplemented with *R. ilealis* showed impaired glucose tolerance (AUC and 15 mins. glucose levels)”. However, no significant difference is shown in the related graph (Fig 3a, last graph).
 - Fig 4e-h, what was the test used? Are the data displayed with SD or SEM? In Fig 4f, the difference between the WD group and the lactobacilli groups is quite low, is that biologically relevant?

Thanks to the reviewer for this important comment. We now included a statistics section in M&M where we described in more details all statistical approaches that have used in the paper and the strategy for choosing statistical tests based on type of data (i.e. data distribution and experimental design). Furthermore, to be specific we added to each figure legend description of the statistical test utilized.

- The authors state that “Recently, there were several reports concerning the effects of pathogens and probiotics on the gut microbiota and a possibility that their effect on the host is mediated by those alterations (27,28). Therefore, we sequenced 16S rRNA gene in ileum and fecal samples from mice supplemented with three candidate bacteria. In contrast to other studies, very few changes were observed in the ileal and stool microbiota composition due to supplementation by these microbes”.
Refs 27 and 28 are quite outdated and several reports have now repeatedly shown that probiotics/ bacteria supplementation, in their large majority, do not affect gut microbiota taxonomic composition (Reviewed in Kristensen, et al, Genome Medicine 2016; also shown in Simon et al Diabetes Care 2015 ; Maldonado-Gomez et al, Cell Host & Microbe 2017; <https://isappscience.org/for-scientists/resources/probiotics/>)

Thanks for pointing out to these papers, which we added. We re-wrote the discussion of this topic according your suggestion pointing to previous studies that agree with our results and few others with opposite data and some strong negative statements about probiotics.

Please see details on potential harmful effects of probiotics below. Both points of view are now presented and discussed in the corresponding section of the main text.

We now see that we did not cite two recent studies that generated a lot of discussions regarding potential harmful effects of probiotics. Both were published in Cell 2018 (PMID 30193112; 30193113) by Elinav&Segal labs, who said “Contrary to the current dogma that probiotics are harmless and benefit everyone, these results reveal a new potential adverse side effect of probiotic use with antibiotics that

might even bring long-term consequences.” Thus this is definitely a topic of active discussions and it warrants future studies.

Reviewer #2 (Remarks to the Author):

The major claim of this manuscript is that microbes interact (positively and negatively) with liver mitochondria in a way that modifies the outcome of T2D-related tests such as glucose intolerance. As per the title, this may mean that Lactobacilli are capable of interacting with liver mitochondria in order to attenuate diabetes induced by a western diet in mice.

Overall, I find this to be interesting proof-of-principle for the effect of microbes from the gut microbiome on the severity of T2D. Given that some of the results appear to have small effect sizes (but have statistical significance), the biggest question in my mind is whether pro or prebiotics will be useful in humans, as the authors imply. That said, I do think that this manuscript will change the thinking in the field and I also think the authors have appropriately limited their conclusions. There is still not a strong consensus or awareness of the links between the microbiome and mitochondria - likely co-evolutionary partners - and the work in this manuscript will add to an important body of literature. More specifically in this work, the authors show that changes in gene expression are affected by alterations in the gut microbiota. This is a hallmark in the sense that this is a study design the probes cause and effect (unlike past studies that looked at genetically modified mouse mitochondria or other heavier handed tweaks). I think the weaker effect size is in that sense realistic.

Thanks so much to the reviewer for the nice comment! Admittedly, we were very surprised by the results we got from the liver gene expression showing massive upregulation of genes involved in mitochondrial functions. We agree that the topic of mitochondria-microbiota interactions is a very fascinating and largely unexplored one.

- The largest change that would make this more convincing would be to add a mechanistic element to the analysis. For example, the mitochondria gene expression correlations in Fig 4b are significant, but somewhat unexplained. This leaves me a little confused as to what to make of these seemingly random mitochondrial functional categories.

A. We have supporting figure 4 that shows a network of up-regulated mitochondrial functions instead of the list, which complements main figure 4. Also, Fig. 4C shows genes organized by the mitochondrial complexes.

B. More importantly, we have performed a new network analysis (Fig. 5a-c), which showed that several genes involved in mitochondrial organization/removal are connected to main glucose metabolism parameters altered by Lactobacilli supplementation (and not to other phenotypes such as body weight and fat amounts). Although, only perturbation experiments (such as KOs) would provide complete evidence of their action, these results are highly suggestive of liver mitochondria at least partial controlling systemic glucose metabolism.

C. The last, but not the least: our new metabolomics investigation (Fig. 5d-f) points to glutathione and bilirubin as two main metabolites that potentially mediate beneficial effects of *Lactobacilli* on liver mitochondria. Besides abundant literature that support role of these two metabolites (that can be the complementary – see PNAS Sedlak et al 2009), we have tested glutathione effects *in vitro* (as it was the

top increased metabolite in SPF supplementation studies) and found that it induces expression of several mitochondrial genes upregulated by *Lactobacilli in vivo* (see details in Fig. 5).

- In addition, the transkingdom network is hard to read. Given the importance and prominence of the transkingdom network in the title, abstract, and text, I would have expected to be able to see the network and understand the connections that the authors draw between OTUs and who is an influencer, improver, etc.,. If there was a better way to make that clear, this would improve the impact of the manuscript enormously. Similarly, it seems like really the understandable element is how the nodes are scored for keystone-ness, but that is also confusing in places. For instance, in Fig 1d, it appears that the species that are ultimately tested are not the 4 most keystone species. If the logic is that keystone-ness led to the future tests, why were some of the more keystone species skipped?

We agree that it would be ideal “to be able to see the network and understand the connections that the authors draw between OTUs and who is an influencer, improver, etc., “

The standard practice in the network analysis and visualization, however, is to generate the network, show it in its complexity and perform the investigation of biologically meaningful topological parameters (in this case Bi-partite betweenness centrality and keystone-ness) and show them in separate graphs (as we did). While it is correct that the whole network is not very interpretable as a static picture, it is a standard visualization expected to be demonstrated as a whole.

To overcome this limitation, it is now possible to upload networks into a public repository for sharing networks (<http://www.ndexbio.org/#/>), which did for both the transkingdom and newly built multi-omic network. This allows not only for full transparency but also for its further exploration not necessary related to our own network interrogations. The links are in data access section of the manuscript and also copied herein.

Transkingdom network access: <https://tinyurl.com/TK-NW-Fig-1C>

Multi-omic network access: <https://tinyurl.com/multi-omic-NW-Fig-5A>

Regarding the keystone-ness:

We decided to analyze keystone-ness because it is highly discussed measure/concept in the microbiome field indicative of microbes that have strong effect of the rest of microbial community.

Our study is investigating which microbes contribute to glucose metabolism in the mammalian host.

There is a large discussion in the field about “bad” and “good” microbiome for the host. Therefore, we wanted to know whether microbes that regulate some host functions are also important for control of the rest of gut microbiota. The analytical answer to this question was NO. In other words, the transkingdom network analysis shows that our top OTUs predicted to control host phenotypes have low keystone-ness, and vice versa, the top keystone-ness OTUs are not predicted to significantly influence the host.

Thus, according to the original goal of the study we chose to test microbes with a potential effect on the host.

Interestingly, though, supplementation with these microbes did not affect the rest of gut microbiota as predicted by their low keystone-ness while producing significant predicted effect on the host.

Overall, I think this manuscript is appropriate for publication and could be strengthened by addressing some of the above points.

Thanks! We will try to address the comments as much as we can.

Reviewer #3 (Remarks to the Author):

The current manuscript investigates the effects of two candidate microbes on glucose homeostasis and links these effects to liver mitochondrial function. The studies are interesting and largely novel.

Major Comments:

1. There is no data linking the changes in glucose and mitochondria to alterations in liver triglycerides or liver histology.

We have constructed a new network linking genes regulated by the lactobacilli species, systemic phenotypes and lipids, which points to key candidate genes most important for driving the observed changes.

2. There is no mechanism to explain why glucose tolerance is altered.

We showed that after administration of *L. gasseri* and *L. johnsonii* to mice on western diet reduced liver and visceral fat accumulation, lowered several hepatic lipids and improved hepatic mitochondrial gene expression. All these events are known to be associated with improved tissue response to glucose and insulin (reviewed in PMID 31802013, 30168804, 31130874) hence resulting in better glucose tolerance.

3. What is the mechanism for the reduction in adiposity despite no change in body mass?

Adipose tissue is relatively light, thus a small change in adiposity would not be reflected by body mass.

Reviewers' Comments:

Reviewer #1:

Remarks to the Author:

I thank the authors for their reply. I would like to congratulate the authors for the new metabolomic study which adds a mechanistic aspect to the manuscript and reinforce the link between lactobacilli and mitochondria.

As kindly requested by the authors, please find below reference to previous work reporting that lactobacilli administration can impact mitochondrial function, beside the reference provided by the authors. Tunapong, Eur J Nutr 2018 and Nakagawa et al, Aging Cell 2016.

Minor points

Legend for Fig 3 was not provided.

Reviewer #2:

Remarks to the Author:

The authors have largely met my concerns. I do feel like there are ways to make network visualization more appealing, but that this does often takes quite a lot of work/time. Offering readers direct access to the network is another way of sharing the insights from the network and is well appreciated.

Reviewer #4:

Remarks to the Author:

- It is not very clear why authors used the term "multi-omic network analysis", where it involves only analysis of transcriptomics data, fatty acid and metabolic parameters data. multi-omics analysis typically involves integrated analysis of more than two omics data, at least.
- Experimental design and details should be provided for the metabolomics analyses i.e., animals, "N" number of biological replicates/experiment, diet type, GF or SPF, probiotic supplementation etc.
- Since this is an untargeted metabolomics analysis, please remove "metabolite quantification" in line 270 and instead mention as "metabolite profiling".
- How much serum sample was used for the metabolite extraction?
- For GF mice, which Lactobacillus strain was used? Why did authors select only one strain?
- What is the reason to pool sera from 2 mice for metabolomics analyses? Why not separately?
- Is measured Glutathione, a reduced or oxidized form? Since it is an unstable molecule, measuring just one or two replicates will not give an authentic data. It is not very clear how authors have statistically evaluated the significance using just duplicate values.
- Authors have validated the glutathione surge observed in metabolomics analysis using an in-vitro culture experiment, but haven't shown the mechanism of action as claimed.
- Are authors able to annotate only 133 metabolites using the public data bases? Authors should present the complete metabolomics raw data including all the measured features.

- The main idea of doing untargeted metabolomics is to identify biomarkers, formulate new hypotheses, thus, should include the whole data set for statistical analysis including measured features. If statistically significant, then try to identify the unknown metabolite(s). Authors should clarify why they have opted to consider only annotated metabolites for the statistical analyses.
- Authors claimed that 13 out of 133 metabolites were increased in GF mice (I assume based on fold changes >2 fold) and further analysed in SPF mice. Did authors check the blank measurements? Are results significant after background correction?
- In order to validate the results using untargeted metabolomics, authors should measure these 13 metabolites using targeted and quantification method using standards, at least for glutathione, bile acids and related metabolites.
- Missing values or zero values are very common in untargeted metabolomics analyses. What kind of data imputation and filtering methods were used?
- Authors should help me to understand how a fold change value of 8403600.906 is obtained for bilirubin, where the peak abundances of GF WD is 0 (zero) and GF WD+Lacto is 8403; and in figure 5D plotted a fold change value of 16777216.
- Similarly, fold change values for glutathione (228891.308) and Tauroursodeoxycholic acid (213194.7213), where the denominator is zero, and not matching with the plotted values in figure 5D; and for 1-Methyladenosine (2.12241E-05) and Succinoadenosine (1.73244E-05), where the numerator is zero. These are clearly technical artefacts. Authors based all their claims based on these results. Authors should work with filtered and clean version of the data after quality checks.
- Authors should clarify how the average fold change values for SPF mice were calculated from the raw data.
- Authors should explain why they have chosen to further analyse with only >2 fold change values but not down regulated metabolites.
- Authors should clarify whether they have measured metabolites of 4 different experiments at 4 different time points. If so, how are the batch correction and peak alignment done?
- Authors should deposit the whole metabolomics data to a public data repository.

REVIEWER COMMENTS

Reviewer #1 (Remarks to the Author):

I thank the authors for their reply. I would like to congratulate the authors for the new metabolomic study which adds a mechanistic aspect to the manuscript and reinforce the link between lactobacilli and mitochondria.

Thanks for your supportive comment.

As kindly requested by the authors, please find below reference to previous work reporting that lactobacilli administration can impact mitochondrial function, beside the reference provided by the authors. Tunapong, Eur J Nutr 2018 and Nakagawa et al, Aging Cell 2016.

We would like to thank the reviewer for the references, which we now added to the discussion.

Minor points

Legend for Fig 3 was not provided.

We apologize for this mistake and added Figure 3 legend.

Reviewer #2 (Remarks to the Author):

The authors have largely met my concerns. I do feel like there are ways to make network visualization more appealing, but that this does often takes quite a lot of work/time. Offering readers direct access to the network is another way of sharing the insights from the network and is well appreciated.

Thanks for your supportive comment.

Reviewer #4 (Remarks to the Author):

- It is not very clear why authors used the term “multi-omic network analysis”, where it involves only analysis of transcriptomics data, fatty acid and metabolic parameters data. multi-omics analysis typically involves integrated analysis of more than two omics data, at least.

The reason to call the this multi-omic network is that it best describes the diverse nature of underlying type of data (i.e. transcriptomics, fatty acid panel representing limited lipidomics, and list of phenotypes frequently called phenomics). Noteworthy, while number of lipids and phenotypes theoretically can be expanded, lipidome and phenome are routinely used terms [Bush Nat Rev Genet 2016; Tabassum Nat Commun 2019] to refer a set of lipids and phenotypes, respectively. Therefore, it makes integration these three “omes” in the one network, a multi-omic network.

However, any suggestion on how to change the term “multi-omic network” that would still include different types of data in the network would be very welcome!

Bush, W., Oetjens, M. & Crawford, D. Unravelling the human genome–phenome relationship using phenome-wide association studies. Nat Rev Genet 17, 129–145 (2016).

<https://doi.org/10.1038/nrg.2015.36>

Tabassum, R., Rämö, J.T., Ripatti, P. et al. Genetic architecture of human plasma lipidome and its link to cardiovascular disease. Nat Commun 10, 4329 (2019). <https://doi.org/10.1038/s41467-019-11954-8>

- Experimental design and details should be provided for the metabolomics analyses i.e., animals, “N” number of biological replicates/experiment, diet type, GF or SPF, probiotic supplementation etc.

Thanks for pointing out to the lack of this important information, which is now described in detail in M&M section of metabolomics.

- Since this is an untargeted metabolomics analysis, please remove “metabolite quantification” in line 270 and instead mention as “metabolite profiling”.

we have replaced “metabolite quantification” with “metabolite profiling”.

- How much serum sample was used for the metabolite extraction?

We added to the methods that 30 uL of serum were used.

- For GF mice, which *Lactobacillus* strain was used? Why did authors select only one strain?

We have done the GF monocolonization with *Lactobacilli gasseri*. Before we received the comments from the previous round of reviews we had already completed a “treatment” experiment with *L. gasseri* (previous two experiments both both bacteria were “prevention” design) that showed that supplementation with this bacterium improved glucose intolerance in the established mouse T2D.

Also, while *L. gasseri* showed similar results to *L. johnsonii* we were not very enthusiastic to expand too far *L. johnsonii* studies (provided by Nestec S.A.) because of prospective limitations of patenting the treatment methodology and implementing it in potential clinical trials.

At that moment, we had only few germfree mice available and proceeded by mono-colonizing them with the *L. gasseri* for which we had more data expecting to further expand this experiment later on.

Then experiments were interrupted by the COVID-19 pandemic. Therefore, while we still could do limited *in vitro* work, we decided to perform metabolomics evaluation of serum samples from 5 treatment groups of SPF mice (*L. gasseri* and *L. johnsonii* included) and the one experiment from GF mice. The objective was to use the GF experiment first to establish which metabolites are increased in the presence of *Lactobacilli* while removing the possible confounding effect of other bacteria, and also reducing the number of hypotheses tested in SPF mice. This TWO STEP approach (first germfree monocolonization and then SPF mice supplementation) might be prone to false negatives but would control for FALSE POSITIVE errors, which was our main priority.

- What is the reason to pool sera from 2 mice for metabolomics analyses? Why not separately?

In short, the goal of this small experiment (germfree mice monocolonization) was to establish a list of identifiable metabolites that would be further verified in a number of independent experiments (SPF mice supplementation for two months) where we observed beneficial effects on metabolism. As described above, with the imminent shutdown because of COVID-19, we opted to do metabolomics in pools by experiment (GF and SPF) to save time and still be able to produce meaningful results.

The lengthy answer is the following: the exact quantification of a metabolite was not the focus of the germfree experiment. Mainly, we wanted to (i) shortlist candidate metabolites produced in the presence of *Lactobacilli* ONLY, (ii) identify the most significantly altered metabolites by supplementation of *Lactobacilli* in presence of other microbiota and (iii) those that can potentially control host gene expression related to mitochondrial health, which was requested in the first round of review.

(i) Several studies have analyzed and supported the use of pools of samples for large scale assays [Kendzierski et al., PNAS, 2005; 10.1073/pnas.0500607102] [Peng et al., BMC Bioinformatics, 2003; 10.1186/1471-2105-4-26]. Indeed, Kendzierski et al. recommend

using pooling when fewer than three assays per group will be performed. Identifying top ranked (in terms of Lacto vs WD fold change) metabolites was the priority.

- (ii) Using multiple independent experiments with 4-6 mice pooled per experiment per group can be equally powerful to identify highly significant differentially abundant metabolites as doing it without pooling, albeit at much lower financial cost and in less time [Kendziorski et al., PNAS, 2005; 10.1073/pnas.0500607102] [Peng et al., BMC Bioinformatics, 2003; 10.1186/1471-2105-4-26].
- (iii) Finally, we validated *in vitro* that the top candidate Glutathione can indeed control host genes involved in mitochondrial processes.

- Is measured Glutathione, a reduced or oxidized form? Since it is an unstable molecule, measuring just one or two replicates will not give an authentic data.

We really appreciate for asking this! We indeed observed an increase specifically in reduced glutathione (now added to the text and figure 5). It is important from biological point of view as we further used reduced glutathione in cell culture experiments.

We now also specifically looked for the results of oxidized glutathione and did not see even a trend to be different, that we now mention in the text.

-It is not very clear how authors have statistically evaluated the significance using just duplicate values.

This seems to be some misunderstanding; statistics was not done using “duplicate values”. The statistical analysis was performed based on the results of pools from 5 supplementation experiments in SPF mice (each pool from 4-6 mice) using metabolites preselected based on fold change in germfree monocolonization. The metabolomics in GF mice served as an initial filter to identify the candidate metabolites produced in the presence of only Lactobacilli. While we focused on these 12 metabolites in the SPF mice, their ranking in GF mice was not considered when analyzing SPF mice. We have clarified this in the text to avoid any such confusion for readers (and reviewers).

- Authors have validated the glutathione surge observed in metabolomics analysis using an *in vitro* culture experiment, but haven't shown the mechanism of action as claimed.

We agree that the full investigation of mechanism was not performed. We believe, however, that our study overall pointed to at least one critical mechanism of the effect of probiotics, namely Lactobacilli supplementation can lead to GSH increase which alters liver gene expression leading to improved mitochondrial health and ultimately ameliorates glucose intolerance.

Furthermore, our follow-up investigation exceeded the expectations of the Editors that did “not expect any follow-up on identified metabolites in vitro” as we performed *in vitro* experiments to ensure that glutathione contributes to this mechanism that connected results of gene expression network analysis with the metabolomics finding.

Further in-depth testing of this mechanism, for example using GSH deficient mice, GSH deficient microbes, GSH depleting drugs, and treatment with GSH, would be a logical future continuation of this research, which is beyond the scope of this paper.

Authors should present the complete metabolomics raw data including all the measured features. The complete metabolomics raw data, including all the measured features has been uploaded in Metabolomics Workbench under accession #ST001436 (see reviewer's access in the answer to the last question). In addition, we are submitting the complete list of annotated compounds within this revised version of our manuscript as supplementary table S14a. We loaded data acquired in both, electrospray ionization (ESI) negative and positive modes. As a summary: serum ESI + : 7,100 features with just MS information; 2,461 features with both MS and MS/MS

information; serum ESI - : 2,141 features with just MS information ; 1,204 features with both MS and MS/MS information.

- Are authors able to annotate only 133 metabolites using the public data bases?

We choose a conservative approach as our first goal was to verify if we can find metabolites relevant to investigated the biological process with the highest level of confidence in our annotations (see more detailed explanation of our motivations in the answer to the next question). For this, we used the features with both MS and MS/MS information to search in the online HMDB (<http://www.hmdb.ca/>) and METLIN (<http://metlin.scripps.edu/>) databases. A metabolite was sieved out when a match with a difference between observed and theoretical mass was less than 10 ppm. Then, the metabolite molecular formula of matched metabolites was further identified by the isotopic distribution measurement. By doing so, the number of annotated compounds with a defined identification was significantly reduced. We chose to increase the confidence of our annotations, rather than increase the number of annotated compounds with a lower level of confidence. All our annotated/known compounds have a match score ranging from 36.1 to 57.8 and an isotope similarity between 67.7-99.1%. This description was added to Methods Section corresponding to metabolomics.

- The main idea of doing untargeted metabolomics is to identify biomarkers, formulate new hypotheses, thus, should include the whole data set for statistical analysis including measured features. If statistically significant, then try to identify the unknown metabolite(s). Authors should clarify why they have opted to consider only annotated metabolites for the statistical analyses.

We agree with reviewer that an overall goal of a new study that employs untargeted metabolomics is to identify known and also novel metabolites that might contribute to the investigated phenomenon.

In our situation, however, the metabolomics analysis was a follow-up study where our goal was to find known metabolites (if possible) that could be readily interpreted in the context of biological system we are investigating. We do have a complete comprehension of the limitations of this “under the lamp post” approach. However, we have found at least one (glutathione) and potentially second (bilirubin) metabolite, which are known to play anti-oxidant and mitochondria improving role. Having obtained these important results, we decided to leave the identification of unknown metabolites (which is a lengthy and expensive endeavor) for future work.

- Authors claimed that 13 out of 133 metabolites were increased in GF mice (I assume based on fold changes >2 fold) and further analysed in SPF mice. Did authors check the blank measurements? Are results significant after background correction?

First, we mentioned in the M&M that “Blanks and QC samples were injected every 4-5 samples to monitor platform stability”. As an example, the blank intensity values for the reduced and the oxidized forms of glutathione were zero, and it can be found in the raw data loaded in the Metabolomics Workbench. We now added the following: “QC consisting of all pooled experimental samples provided a ‘mean’ profile representing all analytes encountered during the analysis. The QC samples were run before and after every four serum samples to ensure system equilibration. Based on the reproducibility of our QC and on the intensity of the CUDA, we can say that the instrument was stable during the full randomized batch, and that intensity differences are due to biological differences and not due to technical variation.”

Second, the two-fold was chosen as a semi-arbitrary threshold to exceed any potential problems with technical variation and to be reflective of a biological signal. Estimating technical variation in our data based on the control (ISTD_CUDA) we observe mean fold change of 1.016345673 with STDEV 0.13525399 making detection of fold change >2 caused by technical variation very unlikely ($p < 0.0001$).

Moreover, the metabolomic analysis in GF mice colonized or not with *Lactobacilli* served only as an initial filter to shortlist candidate metabolites. While we focused on these 12 metabolites in the SPF mice, their ranking in GF mice was not taken into account when analyzing SPF mice.

- In order to validate the results using untargeted metabolomics, authors should measure these 13 metabolites using targeted and quantification method using standards, at least for glutathione, bile acids and related metabolites.

While we appreciate your question it would be a separate study itself, which is not possible in the current conditions. Also, considering our *in vitro* validation experiments and other available information such as effect of microbiota on glutathione (Mol Syst Biol. 2015, doi: 10.15252/msb.20156487) and bilirubin (Scientific Reports 2016, doi: 10.1038/srep20127), we believe that glutathione and bilirubin are robust candidates to be involved in beneficial effects of *Lactobacilli*.

We want to be clear that in this particular study our insight into the wealth of metabolomics data is limited to finding some plausible candidate metabolites involved in beneficial effects of *Lactobacilli* rather than establishing the comprehensive metabolic map underlying interaction between western diet, probiotics, microbiota and mammalian host.

Finally, our conservative approach that led to identification of rather low overall number of known metabolites (n=133) allows to be confident in their identity, especially for most critical ones (GSH and bilirubin). Please see details of their spectra matching in the images below.

REDUCED GLUTATHIONE

☆	Compound ID	Description	Adducts	Formula	Reten	Score	Fragmentation	Mass error	Reten	Isotope similarity	Link	Search Configuration
★	44	Glutathione	M+H	C ₁₀ H ₁₇ N ₃ O ₆ S	A	45.1	A 33.8	-1.38		93.21	no...	A Method: METLIN™ MS/MS Librar
☆	HMDB0000125	Glutathione	M+H	C ₁₀ H ₁₇ N ₃ O ₆ S	B	43.7	B 27	-1.38		93.21	no...	B Method: Progenesis MetaScope.
☆	HMDB0062697	Glutathione(1-)	M+H	C ₁₀ H ₁₇ N ₃ O ₆ S	B	43.1	B 24.1	-1.38		93.21	no...	B Method: Progenesis MetaScope.

OXIDIZED GLUTATHIONE

☆	Compound ID	Description	Adducts	Formula	Reten	Score	Fragmentation score	Mass error (ppm)	Reten	Isotope similarity	Link	Search Configuration
★	HMDB0003337	Oxidized glutathione	M+H, M...	C ₂₀ H ₃₂ N ₆ O ₁₂ S ₂	B	53.6	B 82.3	-1.71		87.66	nonlinea...	B Method: Progenesis MetaScope.
★	45	Glutathione, oxidized	M+H, M...	C ₂₀ H ₃₂ N ₆ O ₁₂ S ₂	A	45.9	A 44.1	-1.71		87.66	no...	A Method: METLIN™ MS/MS Librar

BILIRUBIN

☆	Compound ID	Description	Adducts	Formula	Reten	Score	Fragmentation	Mass error	Reten	Isotope similarity	Link	Search Configuration
★	HMDB0000054	Bilirubin	M+H...	C ₃₁ H ₃₆ N ₄ O ₆	B	51.3	B 67.8	-2.57		91.52	no...	B Method: Progenesis MetaScope. Da
☆	HMDB0000488	4E,15Z-Bilirubin IXa	M+H...	C ₃₁ H ₃₆ N ₄ O ₆	B	51.3	B 67.8	-2.57		91.52	no...	B Method: Progenesis MetaScope. Da
☆	81	Bilirubin	M+H...	C ₃₁ H ₃₆ N ₄ O ₆	A	45.9	A 41.2	-2.57		91.52	no...	A Method: METLIN™ MS/MS Library. F
☆	53677	Pubescenol	M+H...	C ₁₂ H ₂₄ O ₁₀	A	38.5	A 0.000377	-0.28		92.92	no...	A Method: METLIN™ MS/MS Library. F
☆	264307	Gly Phe Asn Thr Phe	M+H...	C ₂₈ H ₃₈ N ₆ O ₈	A	38.4	A 1.42	4.32		95.41	no...	A Method: METLIN™ MS/MS Library. F
☆	212778	Gln Gln Phe Tyr	M+H...	C ₂₈ H ₃₈ N ₆ O ₈	A	38.2	A 0.379	4.32		95.41	no...	A Method: METLIN™ MS/MS Library. F
☆	213063	Gln Gln Tyr Phe	M+H...	C ₂₈ H ₃₈ N ₆ O ₈	A	38.2	A 0.377	4.32		95.41	no...	A Method: METLIN™ MS/MS Library. F

- Missing values or zero values are very common in untargeted metabolomics analyses. What kind of data imputation and filtering methods were used?

Thanks for pointing out to this. You are correct. We initially substituted 0 by 0.001, which resulted in extremely high fold changes. We agree that this randomly selected low value is far from the best option for imputation of Data Below the Quantification Limit. Therefore, we now used another approach. Studies we found that developed methods for "below quantification limit" imputations are dealing with a bit different type of data (i.e. targeted type of assays that have known lower limits). These imputations usually range from using zero values to "lower limit of quantification" (LLOQ) divided by 2 with more sophisticated modeling in between (PMID: 11768292; PMID: 21191855; PMID: 26038706; PMID: 15238313). While we could not implement more sophisticated versions of imputation because we use untargeted metabolomics, we have to avoid using zero values, therefore, we chose to be on conservative side. Specifically, we replaced 0 with "minimal values" calculated as: **3 STDEV of technical variation subtracted from the minimal measured level of a given metabolite in our study.**

Technical variation was defined by using CUDA and corresponded to STDEV of 0.135 with mean 1.02. Accordingly, we recalculated ratios and observed more realistic ratios now (e.g. ~ 10 for glutathione, and ~48 for bilirubin). We modified corresponding figures, text and tables accordingly.

Noteworthy, because the exact value of these ratios are not used in any statistical analysis (only for selection of ratios >2 in germfree colonization experiment) we believe they do not affect any of our conclusions. While it does not change any analytical outcomes, we will be happy to use other data imputation that is considered more appropriate.

- Authors should help me to understand how a fold change value of 8403600.906 is obtained for bilirubin, where the peak abundances of GF WD is 0 (zero) and GF WD+Lacto is 8403; and in figure 5D plotted a fold change value of 16777216.

Previously, we substituted 0 by 0.001 which gave such high numbers.

As we discussed above we now recalculated ratio based on a better way to impute data. Also, usage of log2 transformation for Y axes in Figure 5D misleads about the actual value. We changed the graph and are using log10. We thank the reviewer for bringing this to our attention and help avoid confusion for readers and reviewers.

- Similarly, fold change values for glutathione (228891.308) and Tauroursodeoxycholic acid (213194.7213), where the denominator is zero, and not matching with the plotted values in figure 5D; and for 1-Methyladenosine (2.12241E-05) and Succinoadenosine (1.73244E-05), where the numerator is zero. These are clearly technical artefacts. Authors based all their claims based on these results. Authors should work with filtered and clean version of the data after quality checks.

You are correct that extreme values come from substitution of zero by 0.001. As described above, we changed data imputation for 0 values. Specific reasons why we believe that zeros represent values below sensitivity rather than technical artifacts:

First, for the most important results (i.e. glutathione and bilirubin), there are several publications that microbiota leads for their strong increase in different body locations. For example, there are clear results showing microbiota promoted increase of glutathione in the gut (including our unpublished work using Metabolon platform and a study from another group (Mol Syst Biol. 2015 Oct; 11(10): 834) and bilirubin in serum (Scientific Reports volume 6, Article number: 20127 (2016)).

Second, we manually curated all our annotations to verify the fragmentation pattern and to ensure the highest level of confidence in our annotations. As a result of this manual reanalysis we have found uncertainty in the annotation of the tauroursodeoxycholic acid (found in ESI –

mode), which is coeluting very close to taurodeoxycholic acid (found in ESI + mode). Since we can't reliably annotate this metabolite, it was removed from the list of annotated metabolites.

- Authors should clarify how the average fold change values for SPF mice were calculated from the raw data.

The average fold-change was calculated by dividing the level of a given metabolite in a Lactobacilli-supplemented group by a level of the same metabolite in the corresponding control group and then we averaged these ratios across all five SPF experiments.

- Authors should explain why they have chosen to further analyse with only >2 fold change values but not down regulated metabolites.

The germfree experiment was planned to test a straightforward hypothesis that some metabolites that appear in the presence of Lactobacilli can play the mechanistic role for mitochondrial functions. Therefore, metabolites inhibited by the presence of Lactobacilli were not pertinent for this hypothesis. Theoretically, one can come up with a hypothesis that would require to look into down-regulated metabolites. However, our goal was to optimize the design of the follow-up investigation that we could find the most relevant up-regulated metabolites based on available amount of time and other circumstances (multiple restriction for lab use related to COVID).

- Authors should clarify whether they have measured metabolites of 4 different experiments at 4 different time points. If so, how are the batch correction and peak alignment done?

All the pools of sera were measured in a single fully randomized batch. Therefore, no batch correction was needed. Raw data were imported into Progenesis QI software (Version 2.3, Nonlinear Dynamics, Waters) in order to perform data normalization, feature detection, peak alignment, and peak integration.

- Authors should deposit the whole metabolomics data to a public data repository.

The complete metabolomics raw data, including all the measured features has been loaded in Metabolomics Workbench under accession #ST001436.

Use this link to allow the reviewer access:

<http://dev.metabolomicsworkbench.org:22222/data/DRCCMetadata.php?Mode=Study&StudyID=ST001436&StudyType=MS&ResultType=5&Access=YbbM9471>

Reviewers' Comments:

Reviewer #4:

Remarks to the Author:

Authors have addressed the concerns satisfactorily.

Authors have pointed 2 references for lipids and phenotypes and the correct reference for lipidome is Tabassum Nat Commun 2019, which is a proper lipidomics analysis covering a wide range of lipid molecular species from various lipid classes.

Final comments & answers

Reviewer #4 (Remarks to the Author):

Authors have addressed the concerns satisfactorily.

Authors have pointed 2 references for lipids and phenotypes and the correct reference for lipidome is Tabassum Nat Commun 2019, which is a proper lipidomics analysis covering a wide range of lipid molecular species from various lipid classes.

Response: thanks, we added this reference.